# Stochastic Gradient Methods under Heavy-Tailed Noises in Weakly Convex Optimization

**Tianxi Zhu** [1]   **Yi Xu** [1]   **Qi Wang** [2]   **Xiangyang Ji** [2]

## Abstract

Many empirical studies have shown that the noise distribution of stochastic gradients often exhibits heavy tails when stochastic optimization methods are employed in machine learning tasks. Most existing theoretical analyses of heavy-tailed stochastic methods rely on various convexity and smoothness assumptions and our knowledge of how heavy-tailed stochastic methods behave in the setting of weakly convex optimization is still limited. In the weakly convex setting, this paper derives new upper bounds on the convergence of the stochastic gradient method (SGD) under heavy-tailed noises. In particular, for vanilla SGD, we establish an in-expectation convergence guarantee on the bounded constrained domain under the assumption of bounded $p$-th central moment ($p$-BCM) of the gradient noise, and a high-probability guarantee on the unbounded domain when the noise follows a heavy-tailed sub-Weibull distribution. By equipping SGD with the gradient clipping (Clip-SGD), we demonstrate that it achieves high-probability convergence in the unbounded domain under the $p$-BCM gradient noise. All of our high-probability convergence bounds depend on the failure probability only through polynomial-logarithmic factors. Finally, we present experiments to validate our theoretical findings.

## 1. Introduction

In this paper, we focus on the following stochastic optimization problem, which commonly arises in the field of machine learning.

$$\min_{x \in \mathcal{X}} f(x), \quad f(x) \triangleq \mathbb{E}_{\xi \sim \Xi}[f(x, \xi)] \tag{1}$$

where $\xi$ is a random variable usually following an unknown distribution $\Xi$, $f(x) : \mathbb{R}^d \mapsto \mathbb{R} \cup \{+\infty\}$ is a closed proper function and $\mathcal{X}$ is closed and convex. In particular, this paper focuses on $\rho$-weakly convex objectives, that is, $f$ such that $f + \rho\|x\|^2/2$ is convex for some $\rho > 0$. Weakly convex functions constitute a broad class that includes all convex functions, smooth functions with Lipschitz continuous gradients, and certain compositions of convex functions with smooth functions (Drusvyatskiy & Paquette, 2019; Davis & Drusvyatskiy, 2019). Moreover, many optimization problems arising in machine learning and large-scale optimization are weakly convex optimization problems(Davis et al., 2018).

To address (1), stochastic gradient descent (SGD)[1], which updates the iterate using an unbiased estimator $g(x, \xi)$ of the true subgradient $g(x) \in \partial f(x)$ , has become a widely adopted optimization method in modern machine learning. A standard assumption in much of the existing literature on SGD convergence is that the stochastic noise term $\epsilon(x, \xi) := g(x, \xi) - f(x)$ has bounded variance. Concretely, one assumes either $\sup_{x \in \mathcal{X}} \mathbb{E}_{\xi}\left[\|\epsilon(x, \xi)\|^2\right] \leq \sigma^2$ or $\sup_{x \in \mathcal{X}} \mathbb{E}_{\xi}\left[\exp(\|\epsilon(x,\xi)\|^2/\sigma^2)\right] \leq \exp(1)$ for some constant $\sigma > 0$. The second condition is typically known as the sub-Gaussian assumption.

However, recent studies on training large language models (LLMs) (Ahn et al., 2024), image classification (Battash et al., 2024), and policy optimization in reinforcement learning (RL) (Garg et al., 2021) have empirically found that stochastic gradient noise is non-Gaussian and heavy-tailed, indicating that the bounded variance assumption may be too restrictive. To formalize the heavy-tailed noise, we begin by introducing the following assumption.

**Assumption 1.1.** Suppose that we have access to stochastic subgradients with $\mathbb{E}_{\xi}[g(x, \xi)] \in \partial f(x)$, and define $\epsilon(x, \xi) := g(x, \xi) - \mathbb{E}_{\xi}[g(x, \xi)]$. Assume there exist some $p \in (1, 2]$ and $\sigma > 0$ such that $\epsilon(x, \xi)$ has the $p$-th bounded

[1]School of Control Science and Engineering, Dalian University of Technology, Dalian, China [2]Department of Automation, Tsinghua University, Beijing, China. Correspondence to: Yi Xu <yxu@dlut.edu.cn>.

*Proceedings of the 43rd International Conference on Machine Learning*, Seoul, South Korea. PMLR 306, 2026. Copyright 2026 by the author(s).

---

[1]For simplicity, throughout this paper, we will also refer to projected stochastic subgradient descent as SGD.

central moment ($p$-BCM), i.e.,

$$\sup_{x\in\mathcal{X}} \mathbb{E}_\xi[\|\epsilon(x,\xi)\|^p] \leq \sigma^p. \qquad \text{($p$-BCM)}$$

When $p < 2$ in Assumption 1.1, the variance $\|\epsilon(x,\xi)\|^2$ may be infinite, which prevents the application of conventional convergence analysis techniques. Compared to vanilla SGD, the convergence behavior of stochastic gradient methods that use adaptive step sizes in heavy-tailed settings has been extensively studied. For instance, there are SGD with gradient clipping (Clip-SGD) (Zhang et al., 2020; Sadiev et al., 2023; Nguyen et al., 2023; Liu & Zhou, 2023), SGD with gradient normalization (NSGD) (Hübler et al., 2025; Liu & Zhou, 2025), as well as the combinations of of these methods (Cutkosky & Mehta, 2021; Liu et al., 2023b; Sun et al., 2025). A shared feature of these methods is that their step-size adjustment is governed by the norm of the stochastic gradient.

Despite the substantial theoretical success of these methods, the aforementioned works exhibit certain limitations. For Clip-SGD, the theoretically required clipping threshold must increase over time and depend on $p$. In contrast, in real-world machine learning applications, the clipping threshold used for gradient clipping is typically chosen as a small constant (Touvron et al., 2023; DeepSeek-AI et al., 2025). NSGD without momentum theoretically requires a large batch size to ensure convergence (Hübler et al., 2025). Therefore, it is important to study the convergence of classical stochastic gradient methods without nontrivial modifications such as gradient clipping or normalization. Although there are known counterexamples demonstrating that vanilla SGD can diverge under Assumption 1.1 (Zhang et al., 2020), it can still be effective in practice when the heavy-tailed issue is known. This motivates us to investigate that under what conditions vanilla SGD still can converge with heavy-tailed noise.

For more recently, (Fatkhullin et al., 2025) provides the first analysis of vanilla SGD under Assumption 1.1, establishing convergence guarantees across convex, strongly convex, as well as non-convex but Hölder-smooth settings. Under the same heavy-tailed assumption, (Liu, 2026) provides a sequence of analyses for classical online convex optimization methods without any algorithmic modifications, while (He & Lu, 2025) studies the convergence of the standard stochastic proximal subgradient method (SPGM) and its Nesterov-accelerated variant for convex composite optimization problems. To the best of our knowledge, no existing work examines the convergence of classical algorithms in weakly convex settings with heavy-tailed noise even for vanilla SGD, which leads us to pose the following question:

*Q1: In the setting of weakly convex optimization, under what conditions can vanilla SGD still be guaranteed to*

*converge when the stochastic gradient is $p$-BCM?*

Moreover, we are more concerned with convergence with high probability, rather than convergence in expectation in an average-case sense. Recent theoretical results show that, under Assumption 1.1, high-probability upper bounds for stochastic methods without gradient clipping or normalization in convex or smooth settings exhibit an inverse power dependence on the failure probability $\delta$, rather than a polylogarithmic dependence on $1/\delta$ (Chezhegov et al., 2025; Fatkhullin et al., 2025). Therefore, a natural question we want to study is how heavy the noise can be while still allowing vanilla SGD to admit a high-probability upper bound that depends only logarithmically on $1/\delta$. For this purpose, the paper introduces the following assumption on heavy-tailed noise.

**Assumption 1.2.** Suppose that we have access to stochastic subgradients with $\mathbb{E}_\xi[g(x,\xi)] \in \partial f(x)$, and define the noise $\epsilon(x,\xi) := g(x,\xi) - \mathbb{E}_\xi[g(x,\xi)]$. Assume there exists some $\theta \in [1/2, +\infty)$ and $\sigma > 0$, such that $\epsilon(x,\xi)$ satisfies

$$\sup_{x\in\mathcal{X}} \mathbb{E}_\xi\left[\exp\left\{\left(\|\epsilon(x,\xi)\|/\sigma\right)^{\frac{1}{\theta}}\right\}\right] \leq 2 \qquad \text{(sub-Weibull)}$$

Assumption 1.2 specifies that, for every $x \in \mathcal{X}$, the norm of $\epsilon(x,\xi)$ follows a sub-Weibull distribution with tail parameter $\theta$. Several equivalent characterizations of the sub-Weibull distribution are provided in Theorem 2.1 from (Vladimirova et al., 2020). The sub-Gaussian and sub-exponential distributions (see Proposition 2.7 from (Vershynin, 2018)) correspond to particular instances of the sub-Weibull family with $\theta = 1/2$ and $\theta = 1$, respectively. Furthermore, for $\theta > 1$, the sub-Weibull class strictly generalizes the sub-Gaussian and sub-exponential distributions, and its tails become heavier as $\theta$ increases. We also point out that the sub-Weibull noise condition is in fact stronger than Assumption 1.1.

Notably, (Vladimirova et al., 2019; 2020) demonstrate that imposing a Gaussian prior on the weights of a Bayesian neural network leads to a sub-Weibull distribution over those weights. For the high-probability guarantees of stochastic methods with sub-Weibull noise, (Li & Liu, 2022; Madden & Becker, 2024) give an analysis for vanilla SGD in the smooth setting and (Liu & Zhou, 2024; He & Lu, 2025) analysis the stochastic proximal method in the context of convex composite optimization problems. All of the aforementioned works exhibit only a polylogarithmic dependence on $1/\delta$ in the upper bounds they obtained. However, these studies all assume that the objective function meets certain convexity or smoothness conditions, and to the best of our knowledge, none of the existing related work addresses the setting of weakly convex optimization. This leads us to the following natural question:

*Q2: In the setting of weakly convex optimization, can SGD*

*have a high-probability upper bound that depends polylogarithmically on $1/\delta$ when the stochastic gradient noise is heavy-tailed sub-Weibull distribution?*

Finally, we aim to seek a simple modification to the vanilla SGD so that, under Assumption 1.1, the resulting high-probability upper bound depends only polylogarithmically on $1/\delta$. A lot of prior work has shown that, in convex or smooth settings, the upper bound of SGD with gradient clipping under Assumption 1.1 can depend only polylogarithmically on $1/\delta$. To the best of our knowledge, (Hu et al., 2025) provides the only in-expectation upper bound for Clip-SGD in the weakly convex setting. Nonetheless, our understanding of high-probability guarantees—and specifically their dependence on $\delta$—for Clip-SGD in weakly convex setting remains incomplete. This gap motivates us to ask the following question:

*Q3: In the setting of weakly convex optimization, can Clip-SGD have a high-probability upper bound that depends polylogarithmically on $1/\delta$ when the stochastic gradient noise is still assumed to be $p$-BCM?*

Our work seeks to address the stated questions which remain unresolved in the context of weakly convex optimization. We summarize our main contributions as follows:

1. For minimizing weakly convex and $G$-Lipschitz continuous functions, we establish convergence guarantees for SGD without using gradient clipping, normalization, or any other nontrivial algorithmic modifications under two different heavy-tailed assumptions:

   (a) Assuming the stochastic gradient satisfies the bounded $p$-th central moment ($p$-BCM) assumption, we prove that, when optimizing over compact sets, SGD requires at most $\mathcal{O}\big(G^2\varepsilon^{-4} + \sigma^{\frac{p}{p-1}}\varepsilon^{-\frac{2p}{p-1}}\big)$ stochastic gradient oracle calls to obtain an $\varepsilon$-stationary point (with stationarity measured via the gradient of Moreau envelope) in expectation. This leads to a convergence guarantee for SGD under the $p$-BCM assumption in weakly convex optimization, thereby adressing Q1.

   (b) Assuming the stochastic gradient noise is sub-Weibull (Assumption 1.2), the high-probability complexity of SGD over unbounded domains is at most $\widetilde{\mathcal{O}}\big((\log^{2\theta}(1/\delta)\sigma^2 + G^2)\varepsilon^{-4} + \mathbb{1}_{\theta>1/2} \cdot \sigma^2 G^2 \log^{\max\{2\theta,2\}}(1/\delta)\varepsilon^{-4}\big)$, provided we fix the number of iterations in advance and appropriately tune the step-size. The result exhibits a polylogarithmic dependence on $1/\delta$, thereby addressing Q2.

2. In the weakly convex case, we establish a high-probability complexity bound of $\mathcal{O}(\log^{p/(p-1)}(1/\delta)\varepsilon^{2p/(p-1)})$ for Clip-SGD, which exhibits only a polylogarithmic dependence on $1/\delta$ and thus adderssing Q3. For completeness, we also present an in-expectation convergence guarantee in Appendix E. These convergence guarantees are established for unbounded $\mathcal{X}$ and broaden the theoretical understanding of Clip-SGD in the setting of weakly convex optimization with heavy-tailed noise.

Finally, we note that if the noise variance is bounded, then all of our upper bounds on sample complexity collapse to the best-known complexity of $\mathcal{O}(\varepsilon^{-4})$ achieved by gradient-based methods for weakly convex optimization. We compare our analysis of stochastic first-order methods for weakly convex optimization with several previous works (see Table 1 in Appendix A), focusing on both the assumptions and the resulting sample complexity.

## 2. Related Work

In this section, we discuss some theoretical work related to stochastic gradient methods under heavy tails.

**Classic stochastic methods with heavy tails.** For the sub-Weibull noise, (Madden & Becker, 2024) establish high-probability convergence results for SGD, (Liu et al., 2023a) prove in-expectation convergence for AdaGrad-Norm, and (Liu & Zhou, 2024) analyze the high-probability last-iterate convergence of stochastic Bregman divergence-based mirror descent (SMD) for convex composite optimization problems. Moreover, (Yu et al., 2025) investigate SMD with sub-Weibull noise within the framework of multi-agent distributed optimization. More recently, (He & Lu, 2025) investigates the high-probability convergence of the stochastic proximal gradient method (SPGM), both with and without Nesterov's acceleration, under sub-Weibull noise. In another line of study, (Li & Liu, 2022; Chen et al., 2025) examines how sub-Weibull noise influences the stability and generalization performance of SGD.

For the stochastic gradient noise with a bounded $p$-th (central) moment, (Liu & Zhou, 2024) study the last-iterate convergence of SMD, and their analysis rely on the $(1, p/p-1)$-uniform convexity of the function that generates the Bregman divergence. Recently, (Fatkhullin et al., 2025) show that, under merely convex and strongly convex settings, SGD achieves sample complexities of $\mathcal{O}\big(\varepsilon^{-\frac{p}{p-1}}\big)$ and $\mathcal{O}\big(\varepsilon^{-\frac{p}{2(p-1)}}\big)$, respectively, and the comlpexity of $\mathcal{O}\big(\varepsilon^{-\frac{2p}{p-1}}\big)$ for smooth objectives. They also prove that, in smooth optimization, SGD cannot surpass a complexity lower bound of $\Omega\big(\varepsilon^{-\frac{2}{1-r}} + \varepsilon^{-\frac{p}{r(p-1)}}\big)$, where $r$ denotes the exponent governing the polynomially decaying step sizes. (Liu, 2026) derive new regret bounds for several classical algorithms in online convex optimization. In par-

ticular, they demonstrate that AdaGrad-Norm attains the optimal complexity $\mathcal{O}\big(\varepsilon^{-\frac{p}{p-1}}\big)$ without prior knowledge of the Lipschitz constant, the noise level $\sigma$, and the tail index $p$. In the setting of smooth convex composite optimization, (He & Lu, 2025) shows that SPGM achieves the same complexity order in-expectation.

Nevertheless, the analyses in these aforementioned works rely either on convexity or on smoothness.

**Stochastic methods with gradient clipping under heavy tails.** Gradient clipping is primarily employed to handle situations where the gradient noise has infinite variance, i.e., when $p < 2$ in Assumption 1.1. Existing methods include Clip-SGD (Zhang et al., 2020; Nguyen et al., 2023; Liu & Zhou, 2023), clip-AdaGrad (Chezhegov et al., 2025), and Clip-SGD with normalization and momentum (Cutkosky & Mehta, 2021). In addition, (Armacki et al., 2025) study SGD under a general gradient mapping framework that encompasses clipping. All of these works are developed within convex or smooth frameworks. Recently, (Hu et al., 2025) investigated the in-expectation convergence behavior of Clip-SGD for weakly convex objectives in a distributed optimization framework. However, their results only ensure convergence to a stationary point given suitable choices of the algorithm's hyperparameters, without providing an explicit convergence rate. To the best of our knowledge, there is currently no work that establishes high-probability convergence guarantees for Clip-SGD in weakly convex stochastic optimization.

For the lower bound theory, (Vural et al., 2022) proves a lower bound of $\Omega\big(\varepsilon^{-\frac{p}{p-1}}\big)$ for Lipschitz, convex functions defined on a compact domain. For smooth strongly convex and smooth nonconvex functions, (Zhang et al., 2020) derive lower bounds of $\Omega\big(\varepsilon^{-\frac{p}{2(p-1)}}\big)$ and $\Omega\big(\varepsilon^{-\frac{3p-2}{p-1}}\big)$, respectively. Both of these lower bounds are obtained for the class of stochastic first-order methods. More recently, we noticed that that (Fradin et al., 2026) establishes new lower bounds on the sample complexity of momentum variance reduced NSGD, with and without gradient clipping, under Assumption 1.1. In particular, they derive these lower bounds under weaker smoothness assumptions.

Finally, for completeness, Appendix A reviews several recent advances in stochastic weakly convex optimization.

## 3. Preliminary

In this section, we introduce some of the notation and additional assumptions used in the theoretical analysis, as well as the necessary mathematical background.

**Notations.** For any $a$ and $b$ in $\mathbb{R}$, $a \wedge b := \min\{a, b\}$, $a \vee b := \max\{a, b\}$. Given $x, y \in \mathbb{R}^d$, $\langle x, y \rangle$ represents the standard Euclidean inner product, i.e., $\langle x, y \rangle = x^T y$. In our analysis, $\|\cdot\|$ denotes the $\ell_2$ norm. $\Pi_{\mathcal{X}}(y)$ means projecting $y$ onto a given closed convex set $\mathcal{X}$. $\mathcal{O}(\cdot)$, $\Omega(\cdot)$, and $\Theta(\cdot)$ denote the standard asymptotic notations (Howell, 2008), while $\widetilde{\mathcal{O}}(\cdot)$ ignores polylogarithmic factors. For a proper function $f$, its domain is defined as $\mathrm{dom}(f) := \{x \in \mathbb{R}^d : f(x) < +\infty\}$. The notation $\mathrm{int}(\mathcal{S})$ denotes the interior of the set $\mathcal{S}$. In this paper, the subdifferential set is taken to be the Fréchet sub-differential set (Kruger, 2003), that is, $\partial f(x) := \{g : f(y) - f(x) \geq \langle g, y - x \rangle + o(\|y - x\|), \forall x, y \in \mathrm{dom}(f)\}$. If $f$ is $\rho$-weakly convex, then $\partial f(x)$ is reduces to $\{g : f(y) - f(x) \geq \langle g, y - x \rangle - \frac{\rho}{2}\|y - x\|^2, \forall x, y \in \mathrm{dom}(f)\}$

Apart from the assumptions regarding the heavy-tailed distribution of gradient noise, all subsequent theoretical results in this paper are based on the following assumptions.

**Assumption 3.1.** Let $f : \mathbb{R}^d \to \mathbb{R} \cup \{+\infty\}$ be a $\rho$-weakly convex function with $\rho > 0$, and bounded below on $\mathcal{X}$, i.e., $\inf_{x \in \mathcal{X}} f(x) = f^* > -\infty$.

**Assumption 3.2.** Assume that convex closed set $\mathcal{X}$ lies in the interior of $\mathrm{dom}(f)$, which ensures that $\partial f(x)$ exists for every $x \in \mathcal{X}$. Moreover, $f$ is $G$-Lipschitz on $\mathcal{X}$, i.e., $\exists G > 0$, $\|g\| \leq G$, for any $x \in \mathcal{X}$ and $g \in \partial f(x)$.

The Assumption 3.2 is standard in weakly convex optimization (Davis & Drusvyatskiy, 2018; Li et al., 2019; Hong & Lin, 2025). If, in addition, $\mathcal{X}$ is assumed to be bounded, then Assumption 3.2 is automatically satisfied as a consequence of the following proposition, whose proof is given in Appendix D.1.

**Proposition 3.3.** *Let* $f : \mathbb{R}^d \mapsto \mathbb{R} \cup \{+\infty\}$ *is* $\rho$-*weakly convex and* $\mathcal{X} \in \mathrm{int}(\mathrm{dom}(f))$ *is convex and compact, then* $f$ *is* $G$-*Lipschitz on* $\mathcal{X}$.

**Moreau envelope.** In the weakly convex optimization, defining the stationary point of $f$ is crucial due to its non-smoothness and non-convexity. For a $\rho$-weakly convex function $f$, given any $\widehat{\rho} > \rho$ then the Moreau envelope of $f(x)$ can be defined as follows:

$$f_{1/\widehat{\rho}}(x) := \min_{y \in \mathcal{X}} \left\{ f(y) + \frac{\widehat{\rho}}{2}\|y - x\|^2 \right\} \tag{2}$$

More importantly, we have following inequalities hold

$$\begin{cases} f_{1/\widehat{\rho}}(x) \leq f(x), \\ \nabla f_{1/\widehat{\rho}}(x) = \widehat{\rho}(x - \widehat{x}), \\ \mathrm{dist}(0, \partial f(\widehat{x}) + \mathcal{N}_{\mathcal{X}}(\widehat{x})) \leq \|\nabla f_{1/\widehat{\rho}}(x)\| \end{cases} \tag{3}$$

where $\mathcal{N}_{\mathcal{X}}(x) := \{v \mid \langle v, y - x \rangle \leq 0, \forall y \in \mathcal{X}\}$. Note that if $\|\nabla f_{1/\widehat{\rho}}(x)\| = 0$, then there exists a subgradient $g(x) \in \partial f(x)$ such that $\langle g(x), y - x \rangle \geq 0$ for all $y \in \mathcal{X}$, which implies that $x$ is a stationary point satisfying the

**Algorithm 1** Stochastic Subgradient Descent (SGD)

---

**Input:** Initial point $x_1 \in \mathcal{X}$ and the sequence of learning rates $\{\eta_t\}$

**for** $t = 1$ **to** $T$ **do**

    Sample a stochastic subgradient $g_t$ at $x_t$.

    $x_{t+1} = \Pi_{\mathcal{X}} (x_t - \eta_t g_t)$

**end for**

---

optimality condition. Hence, we adopt the gradient of the Moreau envelope as our convergence measure. In this paper, we define the $\varepsilon$-complexity of a given randomized algorithm $\mathcal{A}$ as the number of stochastic oracle calls required to ensure that $\frac{1}{T} \sum_{t=1}^{T} \mathbb{E} \big[ \| \nabla f_{1/\widehat{\rho}}(x_t) \| \big] \leq \varepsilon$, where $\{x_t\}_{t=1}^{T}$ denotes the sequence of iterates generated by $\mathcal{A}$.

## 4. SGD with Heavy-Tailed noises

In this section, we investigate the convergence of vanilla SGD (Algorithm 1) with Assumption 1.1 and 1.2 in the context of weakly convex optimization. Before presenting our convergence results for the SGD method under heavy-tailed noise, we first revisit the conventional analysis of SGD (and its variants) commonly used in the prior literature such as (Davis & Drusvyatskiy, 2018; Mai & Johansson, 2020; Nazari et al., 2020; Hong & Lin, 2025) in the context of weakly convex optimization, and we discuss why this line of analysis becomes insufficient when the stochastic subgradient satisfies Assumption 1.1.

A key step in these analyses is the use of the *nonexpansiveness* of the projection operator, applied at the very beginning of the argument. Specifically, let $\widehat{x}_t := \arg\min_{x \in \mathcal{X}} \big\{ f(x) + \widehat{\rho}/2 \| x - x_t \|^2 \big\}$, we have the following inequality

$$
\begin{aligned}
f_{1/\widehat{\rho}}(x_{t+1}) &\leq f(\widehat{x}_t) + \frac{\widehat{\rho}}{2} \| \widehat{x}_t - x_{t+1} \|^2 \\
&= f(\widehat{x}_t) + \frac{\widehat{\rho}}{2} \| \Pi_{\mathcal{X}} (\widehat{x}_t) - \Pi_{\mathcal{X}} (x_{t+1}) \|^2 \\
&\leq f(\widehat{x}_t) + \frac{\widehat{\rho}}{2} \| \widehat{x}_t - (x_t - \eta_t g_t) \|^2
\end{aligned}
$$

Then we have

$$
2\eta_t \langle g_t, x_t - \widehat{x}_t \rangle \leq f_{1/\widehat{\rho}}(x_t) - f_{1/\widehat{\rho}}(x_{t+1}) + \eta_t^2 \| g_t \|^2 \tag{4}
$$

If we assume that the stochastic subgradient has bounded variance, then the summation of the expectation of the term $\eta_t^2 \| g_t \|^2$ in (4) can be easily controlled by choosing an appropriate step-size sequence $\eta_t$. In contrast, under Assumption 1.1, the expectation of this term may go infinity when $p < 2$. Moreover, it is essentially impossible to eliminate this term in subsequent analysis, as the nonexpansiveness of the projection operator does not provide enough analytical flexibility to compensate for the infinite variance of $g_t$.

We highlight that our in-expectation convergence analysis of vanilla SGD for weakly convex objectives under Assumption 1.1 relies on the following lemma, which generalizes the well-known result in weakly convex optimization.

**Lemma 4.1.** *Let $f$ is $\rho$-weakly convex and $\{x_t\}$ is the sequence generated by SGD, then for any $\widehat{\rho} > \rho$ and $\alpha \in (1, 2]$, we have*

$$
\begin{aligned}
&(\widehat{\rho} - \rho)\eta_t \left\| \nabla f_{1/\widehat{\rho}}(x_t) \right\|^2 \\
&\leq \widehat{\rho} \left( f_{1/\widehat{\rho}}(x_t) - f_{1/\widehat{\rho}}(x_{t+1}) \right) \\
&+ \widehat{\rho}^2 \left( \eta_t \| \nabla_t \| \right)^2 - \widehat{\rho}^2 \eta_t \langle \epsilon_t, x_t - \widehat{x}_t \rangle \\
&+ \frac{\widehat{\rho}^2 (4\alpha - 4)^{\alpha - 1}}{\alpha^\alpha} \left( \eta_t \| \epsilon_t \| \right)^\alpha \| x_{t+1} - x_t \|^{2 - \alpha}, \quad (5)
\end{aligned}
$$

*where $\nabla_t := \mathbb{E}[g_t | x_t]$ and $\epsilon_t := g_t - \nabla_t$*

Note that when $\alpha = 2$, the inequality (5) reduces to (2.6) in (Davis & Drusvyatskiy, 2018), which is frequently used in the prior literature to analyze the convergence of first-order stochastic methods under weakly convex optimization and the variance-bounded assumption. If we set $\alpha$ as $p$, then the error term $\| \epsilon_t \|^p \| x_{t+1} - x_t \|^{2-p}$ can be readily controlled by the defnition of Assumption 1.1 together with the boundedness of the constraint domain.

Now we present the in-expectation convergence of SGD under weakly convex optimization and Assumption 1.1, assuming that $\mathcal{X}$ is compact. To the best of our knowledge, this is the first convergence result established under such a setting.

**Theorem 4.2.** *Suppose that Assumptions 1.1, 3.1 and 3.2 hold, and that the constraint domain $\mathcal{X}$ is bounded with diameter $D_{\mathcal{X}}$. Let the step-size $\eta_t$ of SGD be $D_{\mathcal{X}}/G\sqrt{t} \wedge D_{\mathcal{X}}/\sigma t^{1/p}$. Then, the $\varepsilon$-complexity of SGD is $\widetilde{\mathcal{O}}\big( G^2 \varepsilon^{-4} + \sigma^{\frac{p}{p-1}} \varepsilon^{-\frac{2p}{p-1}} \big)$.*

As shown in Proposition 3.3, if we assume that $\mathcal{X}$ is bounded, i.e., $\mathcal{X}$ is compact, then Assumption 3.2 is automatically satisfied. Moreover, the advantage of the complexity we obtain in Theorem 4.2 is that it can be decomposed into two parts: a deterministic term $G^2 \varepsilon^{-4}$ and a stochastic term $\sigma^{\frac{p}{p-1}} \varepsilon^{-\frac{2p}{p-1}}$. If $\sigma = 0$, the result reduces to the deterministic case, i.e., $\widetilde{\mathcal{O}}\big( G^2 \varepsilon^{-4} \big)$ (we may define $\frac{D_{\mathcal{X}}}{\sigma t^{1/p}} = +\infty$ when $\sigma = 0$). For $p = 2$, i.e., when the gradient noise has bounded variance, the complexity simplifies to $\widetilde{\mathcal{O}}\big( (G^2 + \sigma^2) \varepsilon^{-4} \big)$, which is consistent with the result in (Davis & Drusvyatskiy, 2018) under the variance-bounded assumption. For $p \in (1, 2)$, the dependence on $\varepsilon$ is sub-optimal compared with the lower bound $\Omega\big( \varepsilon^{-\frac{p}{p-1}} \big)$ in (Vural et al., 2022) for merely convex functions and the lower bound $\Omega\big( \varepsilon^{-\frac{3p-2}{p-1}} \big)$ in (Zhang et al., 2020) for non-convex but smooth functions. Nevertheless, we point

out that the lower bounds provided in (Vural et al., 2022) and (Zhang et al., 2020) are established for a broad class of first-order methods, rather than for specific algorithmic instances. For the vanilla SGD method with the step-size $\eta_t = \eta t^{-r}$, (Fatkhullin et al., 2025) shows that there always exist some convex, $L$-smooth, and $G$-Lipschitz functions, together with a stochastic oracle satisfying Assumption 1.1, such that the $\varepsilon$-complexity is at least $\Omega(\varepsilon^{-\frac{2}{1-r}} + \varepsilon^{\frac{p}{r(p-1)}})$ when the diameter $D_{\mathcal{X}}$ is sufficiently large (see Theorem 5.5 in (Fatkhullin et al., 2025)). When $r = 1/p$, the dominant part of the lower bound reduces to $\Omega(\varepsilon^{-\frac{2p}{p-1}})$, which matches the upper bound $\widetilde{\mathcal{O}}(\varepsilon^{-\frac{2p}{p-1}})$ in Theorem 4.2. Since convex and $L$-smooth minimization are both a special case of weakly convex minimization, the upper bound in Theorem 4.2 is optimal up to polylogarithmic factors.

Although Theorem 4.2 establishes the convergence in expectation of SGD under Assumption 1.1, the prescribed stepsize schedule $\eta_t$ depends on the tail index $p$, which is typically unknown in practice. Fortunately, our analytical framework enables us to choose a stepsize that is independent of $p$. The following corollary shows that SGD still converges under Assumption 1.1, even when $p$ is unknown.

**Corollary 4.3.** *Let the assumptions in Theorem 4.2 still hold. If we fix the number of iterations in* SGD *as $T$ and let $\eta_t \equiv \eta/T^r$, where $r \in (0,1)$ and $\eta = D_{\mathcal{X}}/G \wedge D_{\mathcal{X}}/\sigma$. Then, the $\varepsilon$-complexity of* SGD *is $\mathcal{O}(\varepsilon^{-\frac{2}{1-r}} + G^{\frac{1}{r}} \varepsilon^{-\frac{2}{r}} + \sigma^{\frac{1}{r(p-1)}} \varepsilon^{-\frac{2}{r(p-1)}})$.*

Note that $\eta$ in Corollary 4.3 is required to be $D_{\mathcal{X}}/G \wedge D_{\mathcal{X}}/\sigma$ is just for fine-tuning the order of $G$ and $\sigma$. Theoretically, we allow $\eta$ could be any positive real number in the case of unknowing $G$ and $\sigma$. We may set $r = 1/2$ in Corollary 4.3, which is also a common choice in practice. In this case, the complexity reduces to $\mathcal{O}(G^2 \varepsilon^{-4} + \sigma^{-\frac{2}{p-1}} \varepsilon^{-\frac{4}{p-1}})$, whose stochastic term $\sigma^{-\frac{2}{p-1}} \varepsilon^{-\frac{4}{p-1}}$ is clearly worse than that in Theorem 4.2.

Next, we aim to investigate under what types of heavy-tailed noise the vanilla SGD admits a high-probability convergence guarantee with an $\mathcal{O}(\log(1/\delta))$ dependence, where $\delta$ denotes the allowable failure probability. Under the bounded variance condition, a standard approach in high-probability analyses of SGD is to reformulate the noise assumption in a sub-Gaussian form, which enables the application of Freedman-type concentration inequalities. Analogously, we consider the sub-Gaussian-type transformation $\mathbb{E}_\xi \left[ \exp \left( |\epsilon(x,\xi)|^p / \sigma^p \right) \right] \leq \exp(1)$, which corresponds, up to constants, to assuming sub-Weibull gradient noise as defined in Assumption 1.2. Unfortunately, this transformation is no longer mild when $p < 2$: the sub-Weibull condition is strictly stronger than Assumption 1.1, since Lemma 6 in (Madden & Becker, 2024) shows that any sub-Weibull random variable possesses a finite $p$-th moment for all $p > 0$.

Nevertheless, this reformulation still preserves the heavy-tailed nature of the gradient noise and the following theorem shows SGD can achieve high-probability convergence under Assumption 1.2 even with unbounded $\mathcal{X}$.

**Theorem 4.4.** *Suppose that Assumptions 1.2, 3.1 and 3.2 hold. Let step-size $\eta_t$ of* SGD *be $\gamma/\sqrt{t}$, where $\gamma$ can be any positive number. Then with probability at leat $1 - \delta$, $1/T \sum_{t=1}^{T} \mathbb{E}[\|\nabla f_{1/3\rho}(x_t)\|^2]$ is upper bound by*

$$\mathcal{O}\left( \frac{\Delta_1 + \rho G^2 \log T + \rho \log T \left[ \log(1/\delta) \right]^{2\theta} \sigma^2}{\sqrt{T}} \right.$$
$$\left. + \frac{\mathbb{1}_{\theta > 1/2} \cdot \sigma G \left[ \log(T/\delta) \right]^{(\theta \vee 1) - 1} \log(1/\delta)}{\sqrt{T}} \right), \quad (6)$$

*where $\Delta_1 := f(x_1) - f^*$ and $\mathbb{1}_{\theta > 1/2} := 1$ if $\theta > 1/2$; otherwise, $\mathbb{1}_{\theta > 1/2} := 0$.*

Note that (6) can reduce to the deterministic case, i.e., $O((\Delta_1 + \rho G^2 \log T)/\sqrt{T})$, when $\sigma = 0$. From Theorem 4.4, we can observe that the optimization performance growing worse as the tail parameter $\theta$ in Assumption 1.2 is increasing. When $\theta = 1/2$, the sub-Weibull-type noise reduces to sub-Gaussian noise, then (6) show that the iteration complexity is at most $O\left((\Delta_1 + (\log(1/\delta)\sigma^2 + G^2)\log T)/\sqrt{T}\right)$ and we disregard the $\log(1/\delta)$ and $\log T$, this result aligns with the in-expectation results in (Davis & Drusvyatskiy, 2018). When $\theta \in [1/2, 1]$, the dependence on failure probability $\delta$ in (6) is $\mathcal{O}\left([\log(1/\delta)]^{2\theta}\right)$ and when $\theta > 1$, an additional term appeared in (3), i.e., $[\log(T/\delta)]^{\theta-1} \log(1/\delta)$. However, (6) shows that $\theta$ doesn't low the order $\mathcal{O}(1/\sqrt{T})$.

In Theorem 4.4, we adopt a parameter-free and time-varying step-size. It is natural to consider fixing the time-horizon $T$ and assuming we can access to $\Delta_1$, $G$, $\rho$, $\sigma$ and $\theta$ to refine the upper bound in Theorem 4.4. Corollary 4.5 shows that we can improved the dependency on $\delta$ to $\mathcal{O}(\log(1/\delta))$ for $\theta \in [1/2, 1]$ after fixing $T$ and carefully choosing a parameter-dependent step-size. Notably, the term $[\log(T/\delta)]^{\theta-1} \log(1/\delta)$ in (7) still appears when $\theta > 1$ even if we have fixed $T$.

**Corollary 4.5.** *Let the assumptions in Theorem 4.4 still hold. If we fix the number of iterations in* SGD *as $T$ and let $\eta_t \equiv \Theta\left(\sqrt{\Delta_1/\rho([\log(1/\delta)]^{2\theta} \sigma^2 + G^2)T}\right)$. Then with probability at least $1 - \delta$, $1/T \sum_{t=1}^{T} \mathbb{E}[\|\nabla f_{1/3\rho}(x_t)\|^2]$ is upper bound by*

$$\mathcal{O}\left( \sqrt{\frac{\Delta_1 \rho}{T}} \left( \log^\theta(1/\delta)\sigma + G \right) + \frac{\sigma^2 \log(1/\delta)}{T} \right.$$
$$\left. + \frac{\mathbb{1}_{\theta > 1/2} \cdot \sigma G \left[ \log(T/\delta) \right]^{(\theta \vee 1) - 1} \log(1/\delta)}{\sqrt{T}} \right) \quad (7)$$

---

**Algorithm 2** SGD with Gradient Clipping (Clip-SGD)

---

**Input:** Initial point $x_1 \in \mathcal{X}$, the sequence of learning rates $\{\eta_t\}$ and the sequnce of clipping thresholds $\{\lambda_t\}$
**for** $t = 1$ **to** $T$ **do**
    Sample a stochastic subgradient $g_t$ at $x_t$.
    $g_t^{\text{clip}} = \left( \frac{\lambda_t}{\|g_t\|} \wedge 1 \right) \cdot g_t$
    $x_{t+1} = \Pi_{\mathcal{X}} \left( x_t - \eta_t g_t^{\text{clip}} \right)$
**end for**

---

## 5. Clip-SGD with Heavy-Tailed noises

In this section, we study the convergence of Clip-SGD (Algorithm 2) with Assumption 1.1. The only difference between Clip-SGD and SGD is that we manually constrain the norm of the stochastic subgradients to be below the clipping threshold $\lambda_t$.

The following theorem establishes the high-probability convergence of Clip-SGD under Assumption 1.1 without requiring $\mathcal{X}$ to be bounded.

**Theorem 5.1.** *Suppose that Assumptions 1.1, 3.1 and 3.2 hold. Let $\lambda_t = 2G \vee \sigma t^{1/p}$ and $\eta_t = 1/\lambda_t \wedge 1/G\sqrt{t}$ in Clip-SGD. Then with probability at least $1 - \delta$, $1/T \sum_{t=1}^{T} \|\nabla f_{1/2\rho}(x_t)\|^2$ is upper bound by*

$$\mathcal{O}\left( \left( \Delta_1 + (G + \rho) \log(T/\delta) \right) \left( \frac{G}{\sqrt{T}} \vee \frac{\sigma}{T^{1 - \frac{1}{p}}} \right) \right) \quad (8)$$

*Moreover, Let $\lambda_t \equiv 2G \vee \sigma T^{1/p}$ and $\eta_t \equiv 1/\lambda_t \wedge 1/G\sqrt{T}$ after fixing $T$. We have*

$$\mathcal{O}\left( \left( \Delta_1 + (G + \rho) \log(1/\delta) \right) \left( \frac{G}{\sqrt{T}} \vee \frac{\sigma}{T^{1 - \frac{1}{p}}} \right) \right) \quad (9)$$

From Theorem 5.1, we observe that the convergence rate of Clip-SGD can achieve the same order as that of SGD, i.e., $\mathcal{O}(T^{-\frac{p-1}{p}})$. Inequality (9) shows that the polylogarithmic term can be removed by fixing $T$. Differently, unlike SGD, Clip-SGD attains this rate even when $\mathcal{X}$ is unbounded. This suggests that Clip-SGD has an advantage over SGD when handling constrained optimization problems under heavy-tailed noise.

However, it is important to note that ensuring convergence requires both $\eta_t$ and $\lambda_t$ in Theorem 5.1 to depend on $p$, which is typically unknown in practice. Such dependence also appears in both convex and nonconvex optimization (Liu & Zhou, 2023; Nguyen et al., 2023; Sadiev et al., 2023). Although we choose an idealized $\lambda_t$, the threshold $\lambda_t$ can adapt to $\sigma$. In particular, if we set $\sigma = 0$, then $\lambda_t$ simplifies to $2G$. Hence, by Assumption 3.2, $\lambda_t$ always exceeds the norm of the subgradient, implying that Clip-SGD reduces to GD and achieves the convergence rate $\mathcal{O}(1/\sqrt{T})$.

For completeness, we also present the in-expectation convergence guarantee in Appendix E.

We would like to discuss upper bounds in Theorem 5.1 here further. First, to the best of our knowledge, there is currently no existing work that establishes the minimax lower bound for general first-order stochastic methods in the setting of weakly convex optimization. Therefore, at present we do not know whether the rate $\widetilde{\mathcal{O}}(T^{-\frac{p-1}{p}})$ is tight. Nevertheless, we can illustrate, via a concrete example, that improving upon the rate $\widetilde{\mathcal{O}}(T^{-\frac{p-1}{p}})$ is difficult. Consider the following weakly convex function from (Liao et al., 2026, Example 1):

$$f := \begin{cases} -x^2 + 1, & x \in (-1, -0.5), \\ 3(x + 1)^2, & \text{otherwise.} \end{cases} \quad (10)$$

It is straightforward to verify that $f$ is 2-weakly convex, that $f^* := \inf_{x \in \mathbb{R}^d} f(x) = 0$, and that it satisfies the inequality $f(x) - f^* \leq \text{dist}^2(\partial f(x), 0)$. If we apply Clip-SGD to minimize $f$ over $\mathbb{R}^d$, then Theorem 5.1 yields

$$\begin{aligned} \frac{1}{T} \sum_{t=1}^{T} \mathbb{E}[f(\widehat{x}_t) - f^*] &\leq \frac{1}{T} \sum_{t=1}^{T} \text{dist}^2(\partial f(\widehat{x}_t), 0) \\ &\overset{(3)}{\leq} \frac{1}{T} \sum_{t=1}^{T} \|\nabla f_{1/2\rho}(x_t)\|^2 \\ &\leq \widetilde{\mathcal{O}}(T^{-\frac{p-1}{p}}). \end{aligned} \quad (11)$$

On the other hand, existing results show that, for convex objectives, Clip-SGD cannot achieve a rate better than $\Omega(T^{-\frac{p-1}{p}})$ in terms of the function value gap. Consequently, if the rate $\widetilde{\mathcal{O}}(T^{-\frac{p-1}{p}})$ could be improved, Clip-SGD would converge to an optimal solution of $f$ over $\mathbb{R}^d$ even faster than it does for the general convex objectives. This indicates that the upper bound in Theorem 5.1 is inherently difficult to sharpen.

## 6. Experiments

### 6.1. Numerical experiments using synthetic data

We present some numercial experiments on synthetic data to validate our theoretical findings. Let $R > 0$ is a preset radius of $\ell_\infty$ ball, we considered a constrained weakly convex optimization problem of the form:

$$\min_{x \in \mathcal{X}} F(x) := \frac{1}{m} \sum_{i=1}^{m} \left| \langle a_i, x \rangle^2 - b_i \right|, \quad (12)$$

$$\mathcal{X} := \left\{ x \in \mathbb{R}^d : \|x\|_\infty \leq R \right\}$$

We can rewrite $F(x)$ as a compact form: $\|(A^T x)^2 - b\|_1/m$, where $A := [a_1, \ldots, a_m] \in \mathbb{R}^{d \times m}$, $b := [b_1, \ldots, b_d] \in \mathbb{R}^d$.

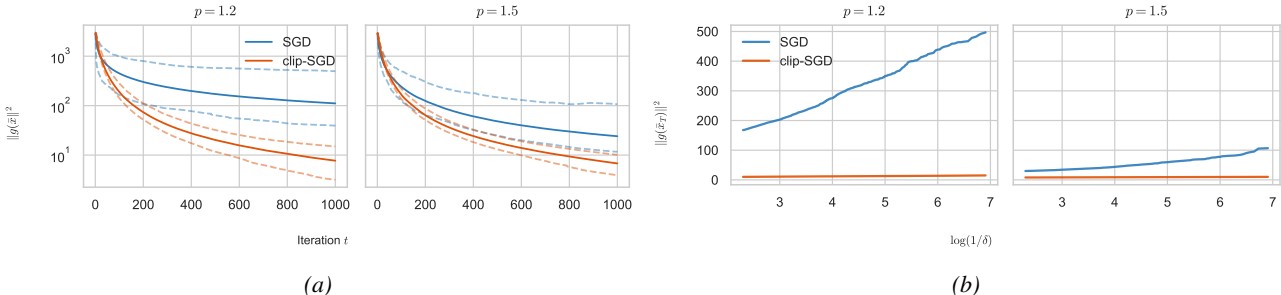

*Figure 1.* High-probability convergence performance of SGD and Clip-SGD under synthetic $p$-BCM noise. **Figure 1a.** The solid lines represent the medians of the runs of $\|g(\bar{x}_t)\|^2$ for both SGD and Clip-SGD, while the dashed lines represent the $\delta$- and $1-\delta$-quantile trajectories. **Figure 1b.** Verifying the $\delta$-dependence of the upper bounds for SGD and Clip-SGD.

Here, we point out $F(x)$ is a loss function comes from a real world problem, i.e., phase retrieval (Duchi & Ruan, 2019). Note that $F$ is the composition of a convex function $\|\cdot\|_1$ and smooth function $(A^T x)^2 - b$ with Lipschitz Jacobian then by Lemma 4.2 from (Drusvyatskiy & Paquette, 2019), $F$ is indeed a nonconvex and nonsmooth but weakly convex function. Moreover, we emulate heavy-tailed gradient noise that satisfies the $p$-BCM or sub-Weibull conditions by adding synthetic noise; further details are provided in Appendix B.1. Let $\bar{x}_t$ denote the average of the first $t$ iterates generated by the algorithms and we use the squared norm of the subgradient at $\bar{x}_t$ as the convergence measure here.

**High-probability convergence performance of SGD and Clip-SGD under the $p$-BCM noise.** We set $R = 10$, take the SGD step size as $\eta_t = 0.05\, t^{-\frac{1}{p}}$, use the clipping threshold for Clip-SGD $\lambda_t = 40 \vee t^{\frac{1}{p}}$, and define the Clip-SGD step size as $\eta_t = \lambda_t^{-1} \wedge 0.05\, t^{-\frac{1}{2}}$. Note that the choices of the order of $\eta_t$ and $\lambda_t$ are aligned with the theoretical selections in Theorem 4.2 and Theorem 5.1. We independently run each algorithms for $10^4$ times with the number of iteration $T = 10^3$. Figure 1a shows the convergence performance by plotting the median, the $\delta$-quantile, and the $(1-\delta)$-quantile of the SGD and Clip-SGD runs, where $\delta = 10^{-3}$. In the theoretical analysis, although we only provide the in-expectation convergence result for SGD, Figure 1a clearly suggests that SGD is much less likely to achieve high-probability convergence compared with Clip-SGD. This is because the $\delta$-quantile curves of SGD deviates significantly from its median curves, whereas the corresponding deviation for Clip-SGD is much smaller.

To further examine the performance differences between Clip-SGD and SGD, we plot the $(1-\delta)$- quantile of each algorithm at $T = 1000$ as a function of $\log(1/\delta)$. Specifically, we sample 1000 values of $\delta$ uniformly from the interval $[10^{-3}, 10^{-1}]$ and record the corresponding pairs $\left(\log(1/\delta), (1-\delta)\text{- quantile}\right)$. Note that, according to Theorem 5.1, the dependence of Clip-SGD on $\log(1/\delta)$ should be at least linear, which is consistent with the empirical

observation in Figure 1b. In contrast, Figure 1b reveals a super-linear dependence on $\log(1/\delta)$ for SGD when $p < 2$, indicating that SGD may fail to achieve high-probability convergence as effectively as Clip-SGD when the noise does not have bounded variance.

Moreover, we conducted additional numerical experiments on synthetic data to corroborate the theoretical results concerning the dependence on the failure probability $\delta$ of SGD under sub-Weibull noise. Owing to space constraints, the full set of experimental results and the complete discussion are deferred to Appendix B.2.

### 6.2. Fine-tune a language model on real data

In this subsection, we present experiments on fine-tuning a pretrained language model to evaluate the practical performance of SGD with momentum (SGDM), Clip-SGD with momentum (Clip-SGDM), and AdamW. We focus on this task because gradient noise with heavy tails has been prominently observed in language modeling tasks (Zhang et al., 2020).

**Experimental Setup** We fine-tune on the SQuAD 2.0 (Rajpurkar et al., 2018) dataset using pretrained BERT$_{\text{BASE}}$ with 12 layers (Devlin et al., 2019). We denote the learning rate (step-size) by $\eta$, the first and second momentum parameters by $(\beta_1, \beta_2)$, the weight decay by $w$, and the gradient clipping threshold by $\lambda$. For SGDM and Clip-SGDM, we set $(\eta, \beta_1, w) = (1e\text{-}2, 0.1, 1e\text{-}5)$, and for Clip-SGDM we set $\lambda = 0.5$. For AdamW, we choose $(\eta, \beta_1, \beta_2, w) = (1e\text{-}5, 0.9, 0.99, 1e\text{-}2)$, matching the training setup as in the BERT paper (Devlin et al., 2019). The batch size is set to 128 for all methods. Moreover, we independently execute each method 5 times using the random seeds $[42, 123, 3407, 2024, 9999]$, and report the average training loss, validation loss, and the corresponding standard deviation.

From Figure 2 and 3, it can be observed that gradient clipping improves the convergence and stability of SGDM in

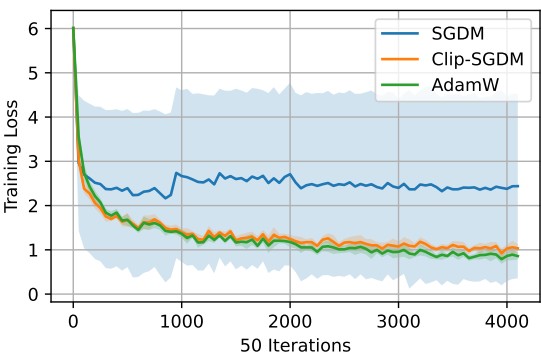

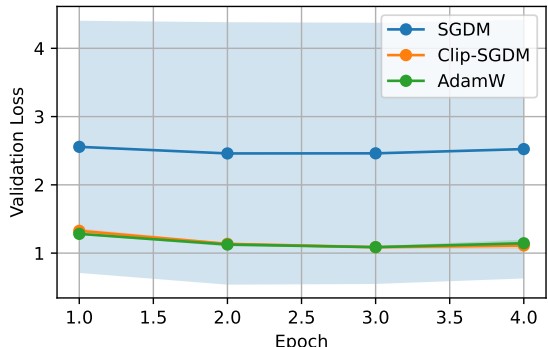

*Figure 2.* The training losses when fine-tuning BERT according to 5 different random seeds. The solid lines represent the average losses, and the width of the shadow region represents the standard deviation.

*Figure 3.* The validation losses when fine-tuning BERT according to 5 different random seeds. The solid lines and the shadow region represent the same meaning as explained in Figure 2.

practice, which aligns with the observation in (Zhang et al., 2020). Here, we point out that although we have theoretically established the convergence of vanilla SGD within bounded regions, SGD can be easily shown to diverge under the $p$-BCM noise assumption on the entire Euclidean space. For example, consider a function $f = x^2/2$ and a stochastic gradient oracle $f'(x, \xi) := x + \xi$, where $\xi \sim \mathcal{D}$ could be any zero-mean distribution with infinite variance, e.g., a two-sided Pareto distribution with tail index $\alpha = 2$. In this case, after one-step SGD update from $x_1 = 0$, we obtain $\mathbb{E}[f(x_2)] = \mathbb{E}[|f'(x_2)|^2/2] = \mathbb{E}[|f'(-\eta_1\xi_1)|^2/2] = \mathbb{E}[\eta_1^2|\xi_1|^2/2] = +\infty$, where the equality holds because we assume that the step-size $\eta_1$ is predefined, deterministic, and non-adaptive. We also can observe that Clip-SGDM, by employing an appropriate clipping threhold $\lambda$, can achieve performance comparable to adaptive method AdamW. This observation reveals that, on the one hand, AdamW itself seems to have a mechanism for gradient clipping and implicitly, adaptively adjusts $\lambda$ during training; on the other hand, although Clip-SGDM can achieve comparable performance to AdamW, this is due to fine-tuning of $\lambda$. Note that the theoretical convergence rate of Clip-SGD in Theorem 5.1 requires $\lambda$ to depend on $p$, which is usually unknown in practice. Therefore, it is urgent to investigate the theoretical convergence guarantee of adaptive methods without gradient clipping under heavy-tailed noises. However, prior to our submission to ICML 2026, to the best of our knowledge, no work has demonstrated that certain adaptive methods without gradient clipping can achieve convergence under non-convex and heavy-tailed noise ($p$-BCM) settings, making this an important research direction for the future.

## 7. Conclusion

We presented a theoretical analysis of stochastic gradient methods under heavy-tailed noise for stochastic constrained weakly convex optimization problems. When the noise is $p$-BCM, we show that vanilla SGD, without adaptive step sizes, still achieves convergence when the constraint domain is bounded. For unbounded domains, we establish convergence guarantees for SGD under sub-Weibull noise and SGD using gradient clipping under $p$-BCM noise. Moreover, our analysis can be generalized to additional optimization settings, including distributed and manifold optimization scenarios.

A limitation of our analysis is that, assuming the stochastic gradient is $p$-BCM, the step sizes of both SGD and Clip-SGD must depend on the typically unknown value of $p$ in order to attain an improved convergence rate. A natural extension of this work is to analyze stochastic gradient methods with heavy-tailed noise in adaptive frameworks, such as AdaGrad, for which existing theory suggests that certain adaptive schemes in the convex settings can be made independent of $p$.

## Acknowledgements

The work is partially supported by National Natural Science Foundation of China under grant number 08120002, Xing Liao Ying Cai Program under grant number XLYC2403174. The authors thank anonymous reviewers for their valuable comments.

## Impact Statement

This paper presents work whose goal is to advance the field of Machine Learning. There are many potential societal consequences of our work, none which we feel must be specifically highlighted here.

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

# Contents

## A. Related Work on Stochastic Weakly Convex Optimization

Stochastic weakly convex optimization have been extensively investigated. (Davis & Drusvyatskiy, 2018) presents the first analysis of SGD and the stochastic proximal gradient method (SPGM) that use the gradient of the Moreau envelope as the convergence measure and establishes a sample complexity of $\mathcal{O}\left(\varepsilon^{-4}\right)$. With the same measure, (Mai & Johansson, 2020) proved, using a Lyapunov-based analysis, that SGD with momentum can achieve the same rate as SGD. (Davis et al., 2018; Liao et al., 2026) show that for certain weakly convex functions satisfying specific regularity conditions, such as sharpness,

PL condition, or quadratic growth—the convergence rate of subgradient methods can be improved to linear.A research direction in weakly convex optimization focuses on the stochastic model-based (SMOD) method(Davis & Drusvyatskiy, 2019; Deng & Gao, 2021; Gao & Deng, 2024). Specially, (Gao & Deng, 2024) relax the $G$-Lipschitz condition in their analysis by exploiting a specific adaptive stepsize scheme together with the local Lipschitz continuity of weakly convex functions. (Nazari et al., 2020; Alacaoglu et al., 2021) established in-expectation convergence guarantees for certain adaptive methods commonly employed in training neural networks. (Hong & Lin, 2025) provides the first high-probability guarantee for AdaGrad-Norm. However, all of these prior arts rely on assumptions of bounded variance or light-tailed noise. We provide a comparison between the theoretical results in this paper and several previous ones for stochastic weakly convex optimization in the following table.

*Table 1.* **Comparison of Prior Analyses of Stochastic First-Order Methods with the Proposed Methods in the Context of Weakly Convex Optimization.** The first column lists various methods: "m-" indicates methods equipped with Polyak momentum. The column "Noise" contains the assumptions imposed on the stochastic gradient noise. Here, "VB" means that the noise variance is bounded. "$\mathcal{X}$" denotes the constraint domain in Problem (1). The column "Grad." specifies whether the norms of the true subgradients are assumed to be uniformly bounded; in other words, whether $f$ is $G$-Lipschitz on $\mathcal{X}$. The final column "Complexity" reports the calls of stochastic oracle to get $\varepsilon$-stationarity in-expectation or with probability at least $1 - \delta$.

| Method | Noise | $\mathcal{X}$ | Grad. | Complexity |
| --- | --- | --- | --- | --- |
| SGD (Davis & Drusvyatskiy, 2018) | VB | Unbounded | Bounded | $\mathcal{O}\left(\varepsilon^{-4}\right)$ |
| m-SGD (Mai & Johansson, 2020) | VB | Unbounded | Bounded | $\mathcal{O}\left(\varepsilon^{-4}\right)$ |
| m-SMOD (Deng & Gao, 2021) | VB | Unbounded | Bounded[a] | $\mathcal{O}\left(\varepsilon^{-4}\right)$ |
| SMOD (Gao & Deng, 2024) | VB | $\mathbb{R}^d$ | Locally bounded[b] | $\mathcal{O}\left(\varepsilon^{-4}\right)$ |
| FEMA (Nazari et al., 2020) | VB | Bounded | Bounded | $\mathcal{O}\left(\varepsilon^{-4}\right)$ |
| AMSGrad (Alacaoglu et al., 2021) | VB | Unbounded | Bounded | $\mathcal{O}\left(\varepsilon^{-4}\right)$ |
| AdaGrad-Norm (Hong & Lin, 2025) | sub-Gaussian | Unbounded | Bounded | $\widetilde{\mathcal{O}}\left(\left[\log(1/\delta)\right]^2 \varepsilon^{-4}\right)$ |
| SGD, Theorem 4.2 | $p$-BCM | Bounded | Bounded | $\widetilde{\mathcal{O}}\left(\varepsilon^{-2p/(p-1)}\right)$ |
| SGD, Theoren 4.4 | sub-Weibull | Unbounded | Bounded | $\widetilde{\mathcal{O}}\left(\left[\log(1/\delta)\right]^{4\theta} \varepsilon^{-4}\right)$ |
| clip-SGD, Theorem 5.1 | $p$-BCM | Unbounded | Bounded | $\widetilde{\mathcal{O}}\left(\left[\log(1/\delta)\right]^{p/p-1} \varepsilon^{-2p/p-1}\right)$ |

[a] Given $\forall y \in \mathcal{X}, \xi \in \Xi$, (Deng & Gao, 2021) require that the stochastic model function $f_y(x, \xi)$ is $L$-Lipschitz continuous for any $x \in \mathcal{X}$. For the stochastic (sub)gradient descent, $f_y(x, \xi) = f(y, \xi) + \langle g(y, \xi), x - y \rangle$. If $\mathcal{X} = \mathbb{R}^d$, then the $L$-Lipschitz continuity sufficiently guarantees $\|g(y, \xi)\| \le L, \forall y \in \mathbb{R}^d, \xi \in \Xi$.

[b] With the two special assumptions are weaker than the global Lipschitz condition on $f$, (Gao & Deng, 2024) shows that for all iteration points $x_t$, the subdifferential $\partial f(x_t)$ is uniformly bounded with high probability.

## B. Additional Experimental Details and Results

### B.1. Synthetic data and noises

For any $x \in \mathcal{X}$, $F$ in (12) has a subgradient $g(x) \in \partial F(x)$ as following

$$g(x) = \frac{2}{m} \sum_{i=1}^{m} \text{sign}(\langle a_i, x \rangle^2 - b_i) \cdot \langle a_i, x \rangle a_i, \tag{13}$$

where

$$\text{sign}(x) = \begin{cases} +1, & x > 0, \\ 0, & x = 0, \\ -1, & x < 0. \end{cases} \tag{14}$$

To simulate the heavy-tailed noises, we independently inject a synthetic noise into the true subgradient $g(x_t)$ of $F$ in each iteration $t$. Specifically, we use the following stochastic gradient to realize SGD and clip-SGD.

$$g_t = g(x_t) + \xi_t \tag{15}$$

where $\xi_t \in \mathbb{R}^d$ is a i.d.d. random vectors generated in the following two ways:

$$\begin{cases} \xi_t = s \odot u^{-1/p}, \\ s \sim \mathsf{Unif}(\{-1, 1\})^d, \qquad (p\text{-BCM}) \\ u \sim \mathsf{Unif}(0, 1)^d. \end{cases} \tag{16}$$

and

$$\begin{cases} \xi_t = \frac{z}{\|z\|} \left[ \log \left( \frac{1}{\sqrt{1-u}} \right) \right]^\theta, \\ z \sim \mathsf{N}(0, 1)^d, \qquad (\text{sub-Weibull}) \\ u \sim \mathsf{Unif}(0, 1)^d. \end{cases} \tag{17}$$

In the first construction of $\xi_t$, its distribution is a two-sided, centered Pareto distribution, which has all moments of order $\alpha \in (1, p)$ finite, while all moments of order greater than or equal to $p$ are infinite. In the second construction of $\xi_t$, the distribution is symmetric with mean zero and is sub-Weibull with parameter $\sigma = 1$.

Note that, as shown in (3), a small value of $\left\| \nabla f_{1/\rho}(\widehat{x}) \right\|$ implies a small gap between $x$ and $\widehat{x}$, as well as a small gap between $\|g(x)\|$ and $\|g(\widehat{x})\|$, since $\|g(x)\|$ is Lipschitz continuous. Therefore, we can use the squared norm of the subgradient at $\bar{x}_t$ as the convergence measure here.

For all experiments, we fix the problem dimension $d$ to 10 and independently generate matrix $A \sim \mathsf{N}(0, 1)^{d \times d}$ and vector $b \sim \mathsf{N}(0, 1)^d$ once, where $\mathsf{N}(0, 1)$ is the standard normal distribution. For the initial point $x_1$ in Algorithms 1 and 2, if the $\mathcal{X}$ is bounded, we choose $x_1$ by sampling uniformly from the boundary of $\mathcal{X}$; otherwise, we randomly draw $x_1 \sim \mathsf{N}(0, 1)^d$.

### B.2. High-probability convergence performance of SGD under synthetic sub-Weibull noises.

We study the convergence behavior of SGD with synthetic sub-Weibull noise when the constraint domain is unbounded, i.e., $\mathcal{X} = \mathbb{R}^d$. From Theorem 4.4, the resulting upper bounds exhibit at most a dependence of either $\mathcal{O}\left( \log(1/\delta)^{2\theta} \right)$ for sufficiently small $\delta$. To verify these theoretical predictions, we assume that the convergence measure scales as $b \log(1/\delta)^a$ and estimate the slope $a$ in the line $a \log \log(1/\delta) + \log b$ using least-squares regression. Specifically, we set the SGD step size as $\eta_t = 0.05 t^{-\frac{1}{2}}$. For each fixed value of $\theta$, we then perform $10^5$ independent SGD runs, each with a fixed number of iterations $T = 10^2$. For $\delta \in [a, b]$, we use the $(1 - \delta)$-quantile $x_{1-\delta}$ to approximate $\mathbb{P}\left\{ \|g(\bar{x}_T)\|^2 \leq x_{1-\delta} \right\} = 1 - \delta$. If our assumption is valid, then the pairs $(\log \log(1/\delta), \log x_{1-\delta})$ should approximately lie on a straight line. To ensure that the pairs corresponding to different $\theta$ all pass through the point $(\log \log(1/b), 0)$, we normalize the quantiles by defining $\widehat{x}_{1-\delta} = x_{1-\delta}/x_{1-b}$. After this normalization, the pairs $(\log \log(1/\delta), \log \widehat{x}_{1-\delta})$ still approximately lie on a straight line with slope $a$. Here we choose three different ranges for $\delta$: $[0.1, 0.2]$, $[0.01, 0.05]$ and $[0.001, 0.005]$.

From Figure 4, it can be observed that the slope $a$ increases as $\theta$ increases and as $\delta$ decreases. It is necessary to exhibit the upper bound on the slope by considering a narrower range of $\delta$. But we need to note that when $\delta$ is extremely small, the $(1 - \delta)$-quantile $x_{1-\delta}$ does not accurately approximate $\mathbb{P}\left\{ \|g(\bar{x}_T)\|^2 \leq x_{1-\delta} \right\} = 1 - \delta$. This issue is evidenced by the increased fluctuations of the curves in Figure 4c compared with those in the other subfigures. Overcoming this limitation would require a substantially larger number of independent runs, exceeding a reasonable computational cost. However, our experimental results show that the empirically estimated value of $a$ is much smaller than the theoretical prediction $2\theta$.

## C. Useful Lemmas

**Additinal notations.** Let $[T]$ be the set $\{1, \ldots, T\}$. We define $\mathcal{F}_t := \sigma(g_1, \ldots, g_t)$ as the filtration generated by Algorithms 1 and 2 up to iteration $t$, where $g_t$ denotes the stochastic subgradient sampled in these algorithms. We let $\mathbb{E}_t[\cdot]$ as the conditional expectaiton $\mathbb{E}[\cdot | \mathcal{F}_{t-1}]$.

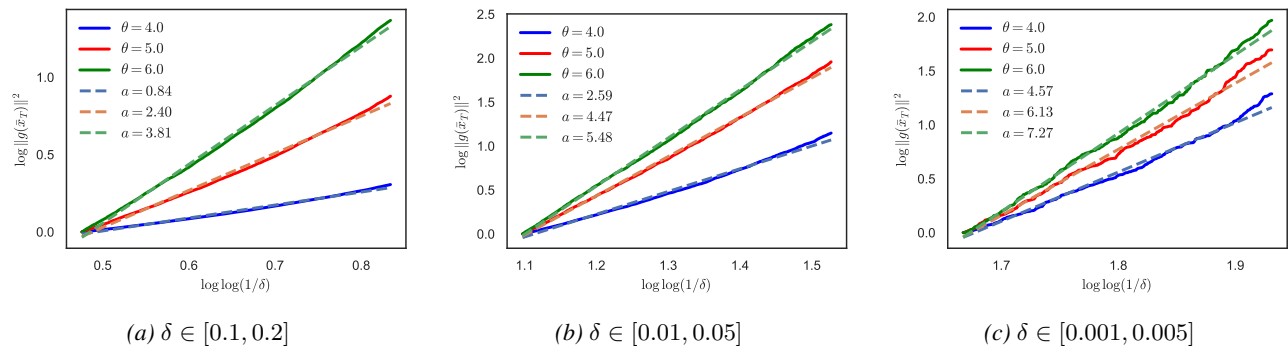

*(a)* $\delta \in [0.1, 0.2]$             *(b)* $\delta \in [0.01, 0.05]$             *(c)* $\delta \in [0.001, 0.005]$

*Figure 4.* The solid lines represent the pairs $(\log\log(1/\delta), \log \widehat{x}_{1-\delta})$, where $\widehat{x}_{1-\delta}$ is the normlized $(1-\delta)$ quantile of observed $\|g(\bar{x}_T)\|^2$ $(T = 100)$ over $10^5$ independent runs. The dashed lines represent the fitted straight lines with slope

**Lemma C.1.** *(Three-point property) Let $\mathcal{S}$ is convex closed and $y = \arg\min_{s \in \mathcal{S}} \{r(s) + B_\psi(s, x)\}$, where $r(\cdot)$ is convex and $B_\psi(\cdot, \cdot)$ is the Bregman divergence defined by 1-strongly convex function $\psi$. Then for any $z \in \mathcal{S}$, we have*

$$r(z) + B_\psi(z, x) \geq r(y) + B_\psi(y, x) + B_\psi(z, y) \tag{18}$$

*Specifically, if $y = \Pi_\mathcal{S}(x - v) = \arg\min_{s \in \mathcal{S}} \left\{ 2\langle s, v \rangle + \|s - x\|^2 \right\}$ and we define $r(s) = 2\langle s, v \rangle$ while taking $B_\psi(x, y)$ to be the Euclidean distance $\|x - y\|^2$, then the inequality becomes*

$$\|z - y\|^2 \leq \|z - x\|^2 - 2\langle v, y - z \rangle - \|x - y\|^2. \tag{19}$$

**Lemma C.2.** *(Theorem 12.17 from (Rockafellar & Wets, 1998)) If a proper function $f : \mathbb{R}^d \mapsto \mathbb{R} \cup \{+\infty\}$ is $\rho$-weakly convex then the following statements are equilvent:*

- *$f(x) + \frac{\rho}{2}\|x\|^2$ is convex on its domain*

- *For any $x, y \in \text{dom}(f)$, the inequality $f(y) \geq f(x) + \langle g(x), y - x \rangle - \frac{\rho}{2}\|y - x\|^2$ holds, where $g(x) \in \partial f(x)$.*

**Lemma C.3.** *For given $\rho$-weakly convex function $f : \mathbb{R}^d \mapsto \mathbb{R} \cup \{+\infty\}$ and $x \in \mathcal{X}$, where $\mathcal{X}$ is convex closed. Let $\widehat{x} := \arg\min_{y \in \mathcal{X}} \{f(y) + \widehat{\rho}/2\|y - x\|^2\}$, where $\widehat{\rho} > \rho$. Then, we have*

$$\|\widehat{x} - x\| \leq \frac{2\|g\|}{\widehat{\rho} - \rho}, \tag{20}$$

*where $g$ can be any subgradient of $\partial f(x)$.*

*Proof.* Note that

$$f(\widehat{x}) + \frac{\widehat{\rho}}{2}\|\widehat{x} - x\|^2 \leq f(x) + \frac{\widehat{\rho}}{2}\|x - x\|^2, \tag{21}$$

then for any $g \in \partial f(x)$, by the equilvent definition of the $\rho$-weakly convexity in Lemma C.2, we have

$$f(\widehat{x}) - f(x) \geq \langle g, \widehat{x} - x \rangle - \frac{\rho}{2}\|\widehat{x} - x\|^2. \tag{22}$$

Combine these two inequalities, we get

$$\frac{\widehat{\rho} - \rho}{2}\|\widehat{x} - x\|^2 \leq \langle g, x - \widehat{x} \rangle \leq \|g\|\|x - \widehat{x}\| \Rightarrow \|\widehat{x} - x\| \leq \frac{2\|g\|}{\widehat{\rho} - \rho} \tag{23}$$

$\square$

Next, we present several concentration inequalities that are employed in our analysis.

**Lemma C.4.** *(Theorem 11 from ([Madden & Becker, 2024](#))) Consider a probability space $(\Omega, \mathcal{F}, \mathbb{P})$, and let $\{\mathcal{F}_t\}$ be the sequence of increasing sub-$\sigma$-algebras of $\mathcal{F}$. Assume there is a martingale difference sequence $\{X_t\}$ adapted to the filtrations $\{\mathcal{F}_t\}$, meaning that $\mathbb{E}[X_t \mid \mathcal{F}_{t-1}] = 0$, and that each $X_t$ is conditionally $\sigma_{t-1}$-sub-Weibull($\theta$) for some $\theta \geq \frac{1}{2}$, i.e.,*

$$\mathbb{E}\left[\exp\left\{(|X_t|/\sigma_{t-1})^{\frac{1}{\theta}}\right\} | \mathcal{F}_{t-1}\right] \leq 2 \tag{24}$$

*where $\sigma_t$ is almost surely nonnegative, $\mathcal{F}_t$-measurable and bounded by $M_{t+1}$. Then, for any $x, \beta \geq 0$ and $\gamma \in (0, 1)$ we have following inequality.*

$$P\left(\bigcup_{\tau \in [T]}\left\{\sum_{t=1}^{\tau} X_t \geq x \text{ and } \sum_{t=1}^{\tau} a\sigma_{t-1}^2 \leq \alpha \sum_{t=1}^{\tau} X_t + \beta\right\}\right) \leq \exp(-\lambda x + 2\lambda^2 \beta) + 2\gamma, \tag{25}$$

*where we require $\alpha \geq b \max_{t \in [T]} M_t$ and $\lambda \in [0, \frac{1}{2\alpha}]$. $a$ and $b$ are defined as following:*

$$a = \begin{cases} 2, & \theta = \frac{1}{2}, \\ (4\theta)^{2\theta} e^2, & \theta \in (\frac{1}{2}, 1], \\ (2^{2\theta+1} + 2)\Gamma(2\theta + 1) + \frac{2^{3\theta}\Gamma(3\theta+1)}{3\log(T/\varepsilon)^{\theta-1}}, & \theta > 1. \end{cases} \quad b = \begin{cases} 0, & \theta = \frac{1}{2} \\ (4\theta)^{\theta}, & \theta \in (\frac{1}{2}, 1] \\ 2\log(T/\varepsilon)^{\theta-1}, & \theta > 1 \end{cases} \tag{26}$$

The following corollary which is easier to use in the analysis can be directly derived from Theorem C.4.

**Corollary C.5.** *Under the same setting in Lemma C.4. With probability at least $1 - \delta$, we have*

$$\sum_{t=1}^{T} X_t \leq 2\alpha \log(2/\delta) + a/\alpha \sum_{t=1}^{T} \sigma_{t-1}^2, \quad \forall \alpha \geq b \max_{t \in [T]} M_t. \tag{27}$$

*Proof.* Let $\beta = 0$

$$
\begin{aligned}
\mathbb{P}\left(\sum_{t=1}^{T} X_t \leq x + \frac{a}{\alpha}\sum_{t=1}^{T}\sigma_{t-1}^2\right) &= 1 - \mathbb{P}\left(\sum_{t=1}^{T} X_t > x + \frac{a}{\alpha}\sum_{t=1}^{T}\sigma_{t-1}^2\right) \\
&\geq 1 - \mathbb{P}\left(\sum_{t=1}^{T} X_t \geq x \text{ and } a\sum_{t=1}^{T}\sigma_{t-1}^2 \leq \alpha\sum_{t=1}^{T} X_t\right) \\
&\geq 1 - \mathbb{P}\left(\bigcup_{\tau \in [T]}\left\{\sum_{t=1}^{\tau} X_t \geq x \text{ and } \sum_{t=1}^{\tau} a\sigma_{t-1}^2 \leq \alpha\sum_{t=1}^{\tau} X_t\right\}\right) \\
&\geq 1 - \exp(-\lambda x) - 2\gamma \tag{28}
\end{aligned}
$$

Let $\lambda = \frac{1}{2\alpha}$, $\varepsilon = \frac{\delta}{4}$ and $x = 2\alpha \log(2/\delta)$, we finish the proof. $\square$

**Lemma C.6.** *(Lemma 16 from ([Madden & Becker, 2024](#))) Given a sequence of random variables $X_1, X_2, \ldots, X_T$. For any $t \in [T]$, $X_t$ is $\sigma_t$-sub-Weibull($\theta$) with $\theta \geq 1/2$, i.e., $\mathbb{E}[\exp\{(|X_t|/\sigma_t)^{1/\theta}\}] \leq 2$. Then, $\forall \gamma \geq 0$, we have*

$$P\left(\left|\sum_{t=1}^{T} X_t\right| \geq \gamma\right) \leq 2\exp\left\{-\left(\frac{\gamma}{v(\theta)\sum_{t=1}^{T}\sigma_t}\right)^{1/\theta}\right\}, \text{ where } v(\theta) = \begin{cases} (4e)^{\theta}, & \theta \leq 1 \\ 2(2e\theta)^{\theta}, & \theta \geq 1 \end{cases}. \tag{29}$$

**Lemma C.7.** *(Bernstein's inequality) Given a probability space $(\Omega, \mathcal{F}, \mathbb{P})$. Let let $\{\mathcal{F}_t\}$ be the sequence of increasing sub-$\sigma$-algebras of $\mathcal{F}$ and $\{X_t\}_{t=1}^{T}$ be a martingale difference sequence adapted to $\{\mathcal{F}_t\}_{t=1}^{T}$, i.e., $\mathbb{E}[X_t|\mathcal{F}_{t-1}] = 0$. Assume that for all $t \geq 1$ we have $|X_t| \leq R$ almost surely, and that $\mathbb{E}[|X_t|^2]$ exists such that $\sum_{t=1}^{T} \mathbb{E}[|X_t|^2 \mid \mathcal{F}_{t-1}] \leq F$. Then with the probability at least $1 - \delta$, we have*

$$\sum_{t=1}^{T} |X_t| \leq \frac{2R\log(2/\delta)}{3} + \sqrt{2F\log(2/\delta)} \tag{30}$$

# D. Missing proofs

### D.1. Proof of Proposition 3.3

*Proof.* Since $f$ is $\rho$-weakly convex, the function $f + \frac{\rho}{2}\|x\|^2$ is convex. By a well-known result from convex analysis, $f + \frac{\rho}{2}\|x\|^2$ is locally Lipschitz continuous on $\mathcal{X}$, which implies that $f$ is also locally Lipschitz continuous on $\mathcal{X}$. Suppose that for any $x \in \mathcal{X}$, $f$ is $\mathsf{L}_x$-Lipschitz continuous on a certain open ball $\mathcal{B}(x)$. By the compactness of $\mathcal{X}$, there exists a finite subcover $\cup_{i \in [I]}\{\mathcal{B}(x_i)\}$ of the open cover $\cup_{x \in \mathcal{X}}\mathcal{B}(x)$. Let $\mathsf{L}_{\mathcal{X}} := \max_{i \in [I]}\mathsf{L}_{x_i}$. For any $x, y \in \mathcal{X}$, the closed segment $l_{x,y} := \{(1-t)x + ty \mid t \in [0,1]\}$ lies in $\mathcal{X}$ due to the convexity of $\mathcal{X}$, and $\cup_{i \in [I]}\mathcal{B}(x_i)$ also covers $l_{x,y}$. The *Lebesgue number lemma* (see Lemma 27.5 from (Munkres, 2000)) ensures the existence of $\delta > 0$ such that for any points $z_i, z_{i+1} \in \{x = z_1, \ldots, z_n = y\} \subseteq l_{x,y}$ with $|z_i - z_{i+1}| < \delta$, there exists $\mathcal{B}_{i,i+1} \subseteq \cup_{i \in [I]}\mathcal{B}(x_i)$ covering the segment $l_{z_i, z_{i+1}}$. Then we have $|f(y) - f(x)| \leq \sum_{i=1}^n |f(z_{i+1}) - f(z_i)| \leq \mathsf{L}_{\mathcal{X}}\sum_{i=1}^n \|z_{i+1} - z_i\| = \mathsf{L}_{\mathcal{X}}\|y - x\|$, which implies that $f$ is $\mathsf{L}_{\mathcal{X}}$-Lipschitz continuous on $\mathcal{X}$. Therefore, $\partial f(x)$ can be uniformly bounded due to the compactness of $\mathcal{X}$. □

### D.2. Proof of Lemma 4.1

*Proof.* Start with the definition of Moreau envelope. function of $f$:

$$f_{1/\widehat{\rho}}(x_{t+1}) = f(\widehat{x}_{t+1}) + \frac{\widehat{\rho}}{2}\|\widehat{x}_{t+1} - x_{t+1}\|^2 \leq f(\widehat{x}_t) + \frac{\widehat{\rho}}{2}\|\widehat{x}_t - x_{t+1}\|^2$$

Let $z$, $x$, $v$ and $y$ in (19) be taken as $\widehat{x}_t$, $x_t$, $\eta_t g_t$ and $x_{t+1}$ respectively, then we get

$$\begin{aligned}
f_{1/\widehat{\rho}}(x_{t+1}) &\leq f(\widehat{x}_t) + \frac{\widehat{\rho}}{2}\left(\|\widehat{x}_t - x_t\|^2 - 2\eta_t\langle g_t, x_{t+1} - \widehat{x}_t\rangle - \|x_t - x_{t+1}\|^2\right) \\
&= \left(f(\widehat{x}_t) + \frac{\widehat{\rho}}{2}\|\widehat{x}_t - x_t\|^2\right) - \widehat{\rho}\eta_t\langle g_t, x_{t+1} - \widehat{x}_t\rangle - \frac{\widehat{\rho}}{2}\|x_t - x_{t+1}\|^2 \\
&= f_{1/\widehat{\rho}}(x_t) - \frac{\widehat{\rho}}{2}\|x_t - x_{t+1}\|^2 + \widehat{\rho}\eta_t\langle\epsilon_t, \widehat{x}_t - x_t\rangle \\
&\quad + \widehat{\rho}\eta_t\left(\underbrace{\langle\nabla_t, \widehat{x}_t - x_t\rangle}_{\text{①}} + \underbrace{\langle\nabla_t, x_t - x_{t+1}\rangle}_{\text{②}} + \underbrace{\langle\epsilon_t, x_t - x_{t+1}\rangle}_{\text{③}}\right)
\end{aligned} \tag{31}$$

For the term ①:

$$\begin{aligned}
\text{①} &\overset{\text{Lemma C.2}}{\leq} f(\widehat{x}_t) - f(x_t) + \frac{\rho}{2}\|\widehat{x}_t - x_t\|^2 \\
&= -\left[\left(f(x_t) - \frac{\widehat{\rho}}{2}\|x_t - x_t\|^2\right) - \left(f(\widehat{x}_t) + \frac{\widehat{\rho}}{2}\|x_t - \widehat{x}_t\|^2\right) + \frac{\widehat{\rho} - \rho}{2}\|x_t - \widehat{x}_t\|^2\right]
\end{aligned} \tag{32}$$

Let $\phi := f(x) - (\widehat{\rho}/2)\|x - x_t\|^2$, which is $(\widehat{\rho} - \rho)$-strongly convex. Thus, for any $\nabla\phi(\widehat{x}_t) \in \partial\phi(\widehat{x}_t)$, we have

$$\phi(x_t) - \phi(\widehat{x}_t) \geq \langle\nabla\phi(\widehat{x}_t), x_t - \widehat{x}_t\rangle + \frac{\widehat{\rho} - \rho}{2}\|x_t - \widehat{x}_t\|^2 \geq \frac{\widehat{\rho} - \rho}{2}\|x_t - \widehat{x}_t\|^2 \tag{33}$$

Note that since $\widehat{x}_t$ is the optimizer of $\phi$ over $X$, the therm of the inner product should be nonnegative by the opimality condition. Now, we can view the term ① as the descent term:

$$\text{①} \leq -(\widehat{\rho} - \rho)\|x_t - \widehat{x}_t\|^2 = -\frac{(\widehat{\rho} - \rho)}{\widehat{\rho}^2}\left\|\nabla f_{1/\widehat{\rho}}(x_t)\right\|^2 \tag{34}$$

For the term ②

$$\text{②} \leq \|\nabla_t\|\|x_t - x_{t+1}\| \leq \eta_t\|\nabla_t\|^2 + \frac{\|x_{t+1} - x_t\|^2}{4\eta_t} \tag{35}$$

Given $\alpha \in (1, 2]$ and $\beta \geq 2$, we have

$$
\begin{aligned}
③ &\leq \|\epsilon_t\| \|x_{t+1} - x_t\| \\
&= \|\epsilon_t\| \|x_{t+1} - x_t\|^{1-2/\beta} \|x_{t+1} - x_t\|^{2/\beta} \\
&= \left(\frac{4\eta_t}{\beta}\right)^{1/\beta} \|\epsilon_t\| \|x_{t+1} - x_t\|^{1-2/\beta} \left(\frac{\beta \|x_{t+1} - x_t\|^2}{4\eta_t}\right)^{1/\beta} \\
&= \left[\left(\frac{4\eta_t}{\beta}\right)^{\alpha/\beta} \|\epsilon\|^\alpha \|x_{t+1} - x_t\|^{\alpha-2\alpha/\beta}\right]^{1/\alpha} \left[\left(\frac{\beta \|x_{t+1} - x_t\|^2}{4\eta_t}\right)^{\alpha/\beta(\alpha-1)}\right]^{(\alpha-1)/\alpha} \\
&\leq \frac{1}{\alpha}\left(\frac{4\eta_t}{\beta}\right)^{\alpha/\beta} \|\epsilon\|^\alpha \|x_{t+1} - x_t\|^{\alpha-2\alpha/\beta} + \frac{\alpha-1}{\alpha}\left(\frac{\beta \|x_{t+1} - x_t\|^2}{4\eta_t}\right)^{\alpha/\beta(\alpha-1)} \quad (36)
\end{aligned}
$$

where the last inequality holds because we use the *Young's inequality*, i.e., $(a^p)^{1/p}(b^q)^{1/q} \leq a^p + b^q$, where $1 < p < +\infty$ and $1/p + 1/q = 1$. Let $\beta = \alpha/(\alpha - 1)$ to make sure $\alpha/\beta(\alpha - 1) = 1$, then we have

$$
③ \leq \frac{(4\alpha - 4)^{\alpha-1}}{\alpha^\alpha} \eta_t^{\alpha-1} \|\epsilon_t\|^\alpha \|x_{t+1} - x_t\|^{2-\alpha} + \frac{\|x_{t+1} - x_t\|^2}{4\eta_t} \quad (37)
$$

Substitute the upper bounds of ①, ② and ③ into the inequality (31) and then multuply $1/\widehat{\rho}$ on the both sides and rearange the terms, we finish the proof. □

### D.3. Proof of Theorem 4.2

*Proof.* By the assumption, $\|\nabla_t\|^2 \leq G^2$ and $\|x_{t+1} - x_t\| \leq D_{\mathcal{X}}$. Then, let $\alpha$ in Lemma 4.1 be $p$ and we have

$$
\frac{\widehat{\rho} - \rho}{\widehat{\rho}^2} \eta_t \|\nabla f_{1/\widehat{\rho}}(x_t)\|^2 \leq \frac{1}{\widehat{\rho}}\left(f_{1/\widehat{\rho}}(x_t) - f_{1/\widehat{\rho}}(x_{t+1})\right) - \eta_t \langle\epsilon_t, x_t - \widehat{x}_t\rangle + (\eta_t G)^2 + \frac{(4p-4)^{p-1}}{p^p}(\eta_t \|\epsilon\|)^p D_{\mathcal{X}}^{2-p}. \quad (38)
$$

Notice that

$$
\begin{aligned}
\mathbb{E}\left[\langle\epsilon_t, x_t - \widehat{x}_t\rangle\right] &= \mathbb{E}\left[\mathbb{E}_t\left[\langle\epsilon_t, x_t - \widehat{x}_t\rangle\right]\right] = 0, \\
\mathbb{E}\left[\|\epsilon_t\|^p\right] &= \mathbb{E}\left[\mathbb{E}_t\left[\|\epsilon_t\|^p\right]\right] \leq \sigma^p
\end{aligned}
\quad (39)
$$

Take the expectation on both sides, we can get

$$
\frac{\widehat{\rho} - \rho}{\widehat{\rho}^2} \eta_t \mathbb{E}\left[\|\nabla f_{1/\widehat{\rho}}(x_t)\|^2\right] \leq \frac{1}{\widehat{\rho}} \mathbb{E}\left[f_{1/\widehat{\rho}}(x_t) - f_{1/\widehat{\rho}}(x_{t+1})\right] + G^2 \eta_t^2 + \frac{(4p-4)^{p-1}}{p^p} D_{\mathcal{X}}^{2-p} \sigma^p \eta_t^p \quad (40)
$$

Take the summation from $t = 1$ to $t = T$, we have

$$
\frac{\widehat{\rho} - \rho}{\widehat{\rho}^2} \sum_{t=1}^{T} \eta_t \mathbb{E}\left[\|\nabla f_{1/\widehat{\rho}}(x_t)\|^2\right] \leq \frac{\Delta_1}{\widehat{\rho}} + \sum_{t=1}^{T} G^2 \eta_t^2 + \sum_{t=1}^{T} \frac{(4p-4)^{p-1}}{p^p} D_{\mathcal{X}}^{2-p} \sigma^p \eta_t^p \quad (41)
$$

Note that $\sum_{t=1}^{T} f_{1/\widehat{\rho}}(x_t) - f_{1/\widehat{\rho}}(x_{t+1}) = f_{1/\widehat{\rho}}(x_1) - f_{1/\widehat{\rho}}(x_{T+1}) = f_{1/\widehat{\rho}}(x_1) - f^* - (f_{1/\widehat{\rho}}(x_{T+1}) - f^*) \leq f_{1/\widehat{\rho}}(x_1) - f^* \leq f(x_1) - f^* =: \Delta_1$. If we choose $\eta_t = \min\left\{\frac{D_{\mathcal{X}}}{G\sqrt{T}}, \frac{D_{\mathcal{X}}}{\sigma T^{1/p}}\right\}$, the right hand side of above inequality can be upper bounded as following

$$
\text{R.H.S.} \leq \frac{\Delta_0}{\widehat{\rho}} + \sum_{t=1}^{T} \frac{D_{\mathcal{X}}^2}{T} + \sum_{t=1}^{T} \frac{(4p-4)^{p-1} p^{-p} D_{\mathcal{X}}^2}{T} \leq \frac{\Delta_0}{\widehat{\rho}} + \left[D_{\mathcal{X}}^2 + (4p-4)^{p-1} p^{-p} D_{\mathcal{X}}^2\right] \log(eT) \quad (42)
$$

and for the left hand side

$$\text{L.H.S.} \geq \frac{\widehat{\rho} - \rho}{\widehat{\rho}^2} \eta_T \sum_{t=1}^{T} \mathbb{E}\left[\left\|\nabla f_{1/\widehat{\rho}}(x_t)\right\|^2\right] \tag{43}$$

Finally, multiply $(\eta_T T)^{-1} = \max\left\{\frac{G}{D_{\mathcal{X}}\sqrt{T}}, \frac{\sigma}{D_{\mathcal{X}} T^{1-1/p}}\right\}$ on the both sides, we get

$$\frac{\widehat{\rho} - \rho}{\widehat{\rho}^2} \cdot \frac{1}{T} \sum_{t=1}^{T} \mathbb{E}\left[\left\|\nabla f_{1/\widehat{\rho}}(x_t)\right\|^2\right] \leq \left\{\frac{\Delta_1}{\widehat{\rho} D_{\mathcal{X}}} + 4\rho \left(\frac{(4p-4)^{p-1}}{p^p} + 1\right) D_{\mathcal{X}} \log(eT)\right\} \cdot \max\left\{\frac{G}{\sqrt{T}}, \frac{\sigma}{T^{1-1/p}}\right\} \tag{44}$$

Let $\widehat{\rho} = 2\rho$, we get

$$\frac{1}{T} \sum_{t=1}^{T} \mathbb{E}\left[\left\|\nabla f_{1/\widehat{\rho}}(x_t)\right\|^2\right] \leq \left\{\frac{2\Delta_1}{D_{\mathcal{X}}} + 4\rho \left(\frac{(4p-4)^{p-1}}{p^p} + 1\right) D_{\mathcal{X}} \log(eT)\right\} \cdot \max\left\{\frac{G}{\sqrt{T}}, \frac{\sigma}{T^{1-1/p}}\right\} \tag{45}$$

$\square$

### D.4. Proof of Corollary 4.3

*Proof.* After fixing the step-size $\eta_t$ as $\widehat{\eta}$, (41) reduces to

$$\frac{\widehat{\rho} - \rho}{\widehat{\rho}^2} \widehat{\eta} \sum_{t=1}^{T} \mathbb{E}\left[\left\|\nabla f_{1/\widehat{\rho}}(x_t)\right\|^2\right] \leq \frac{\Delta_1}{\widehat{\rho}} + G^2 T \widehat{\eta}^2 + \sum_{t=1}^{T} \frac{(4p-4)^{p-1}}{p^p} D_{\mathcal{X}}^{2-p} \sigma^p T \widehat{\eta}^p \tag{46}$$

Let $\widehat{\eta} = \eta/T^r$, where $r \in (0,1)$ and $\eta = \min\{D_{\mathcal{X}}/G, D_{\mathcal{X}}/\sigma\}$. Then, by multiplying $(\widehat{\eta} T)^{-1}$ on both sides, we obtain.

$$\frac{\widehat{\rho} - \rho}{\widehat{\rho}^2} \frac{1}{T} \sum_{t=1}^{T} \mathbb{E}\left[\left\|\nabla f_{1/\widehat{\rho}}(x_t)\right\|^2\right] \leq \frac{(G+\sigma)\Delta_1}{\widehat{\rho} D_{\mathcal{X}} T^{1-r}} + D_{\mathcal{X}} \left(\frac{G}{T^r} + \frac{(4p-4)^{p-1}}{p^p} \frac{\sigma}{T^{r(p-1)}}\right) \tag{47}$$

Let $\widehat{\rho} = 2\rho$, we get

$$\frac{1}{T} \sum_{t=1}^{T} \mathbb{E}\left[\left\|\nabla f_{1/\widehat{\rho}}(x_t)\right\|^2\right] \leq \frac{2\Delta_1}{D_{\mathcal{X}} T^{1-r}} + 4\rho D_{\mathcal{X}} \left(\frac{G}{T^r} + \frac{(4p-4)^{p-1}}{p^p} \frac{\sigma}{T^{r(p-1)}}\right) \tag{48}$$

$\square$

### D.5. Proof of Thereom 4.4

We set $\alpha = 2$ in Lemma 4.1 and get

$$\frac{\widehat{\rho} - \rho}{\widehat{\rho}} \eta_t \left\|\nabla f_{1/\widehat{\rho}}(x_t)\right\|^2 \leq f_{1/\widehat{\rho}}(x_t) - f_{1/\widehat{\rho}}(x_{t+1}) + \widehat{\rho}\eta_t \langle \epsilon_t, \widehat{x}_t - x_t \rangle + \widehat{\rho}\eta_t^2 \left(\left\|\nabla_t\right\|^2 + \left\|\epsilon_t\right\|^2\right) \tag{49}$$

Summing both sides from $t = 1$ to $T$, we get

$$\frac{\widehat{\rho} - \rho}{\widehat{\rho}} \sum_{t=1}^{T} \eta_t \left\|\nabla f_{1/\widehat{\rho}}(x_t)\right\|^2 \leq \Delta_1 + \sum_{t=1}^{T} \widehat{\rho}\eta_t \langle \epsilon_t, \widehat{x}_t - x_t \rangle + \sum_{t=1}^{T} \widehat{\rho}\eta_t^2 G^2 + \sum_{t=1}^{T} \eta_t^2 \widehat{\rho} \left\|\epsilon_t\right\|^2 \tag{50}$$

Observe that for the term $\widehat{\rho}\eta_t^2 \left\|\epsilon_t\right\|^2$, we obtain

$$\mathbb{E}_t\left[\exp\left\{\left(\frac{\widehat{\rho}\eta_t^2 \left\|\epsilon_t\right\|^2}{\widehat{\rho}\eta_t^2 \sigma^2}\right)^{\frac{1}{2\theta}}\right\}\right] = \mathbb{E}_t\left[\exp\left\{\left(\frac{\left\|\epsilon_t\right\|}{\sigma}\right)^{\frac{1}{\theta}}\right\}\right] \leq 2 \tag{51}$$

After taking the full expectation on the both sides, the inequality is still hold. Hence, $\widehat{\rho}\eta_t^2 \|\epsilon_t\|^2$ is $\widehat{\rho}\eta_t^2\sigma^2$-sub-Weibull($2\theta$). Applying Lemma C.6, we obtain, with probability at least $1 - \delta/2$, that

$$\sum_{t=1}^{T} \widehat{\rho}\eta_t^2 \|\epsilon_t\|^2 \leq 2\widehat{\rho}\sigma^2(4e\theta)^{2\theta} \log^{2\theta}(4/\delta) \sum_{t=1}^{T} \eta_t^2 \leq 2\gamma^2\widehat{\rho}\sigma^2(4e\theta)^{2\theta} \log^{2\theta}(4/\delta) \log(eT) \tag{52}$$

If we define

$$\sigma_{t-1} := \widehat{\rho}\eta_t\|\widehat{x}_t - x_t\|\sigma; \quad M_t := \frac{2\widehat{\rho}\eta_t\sigma G}{\widehat{\rho} - \rho}, \tag{53}$$

then we have

$$\mathbb{E}_t\left[\exp\left\{\left(\frac{\widehat{\rho}\eta_t\langle\epsilon_t, \widehat{x}_t - x_t\rangle}{\sigma_{t-1}}\right)^{\frac{1}{\theta}}\right\}\right] \leq \mathbb{E}_t\left[\exp\left\{\left(\frac{\widehat{\rho}\eta_t\|\epsilon_t\|\|\widehat{x}_t - x_t\|}{\sigma_{t-1}}\right)^{\frac{1}{\theta}}\right\}\right] = \mathbb{E}_t\left[\exp\left\{\left(\frac{\|\epsilon_t\|}{\sigma}\right)^{\frac{1}{\theta}}\right\}\right] \leq 2, \tag{54}$$

which indicates that $\{\widehat{\rho}\eta_t\langle\epsilon_t, \widehat{x}_t - x_t\rangle\}_{t\in[T]}$ forms a conditionally $\sigma_{t-1}$-sub-Weibull($\theta$) martingale difference sequence. Note that $\sigma_{t-1}$ is $\mathcal{F}_{t-1}$-measurable and is bounded by $M_t$, thus we can apply Lemma C.4 to get

$$\sum_{t=1}^{T} \widehat{\rho}\eta_t\langle\widehat{x}_t - x_t, \epsilon_t\rangle \leq 2\alpha\log(4/\delta) + \frac{a}{\alpha}\sum_{t=1}^{T} \sigma^2\eta_t^2\widehat{\rho}^2 \|\widehat{x}_t - x_t\|^2 = 2\alpha\log(4/\delta) + \frac{a}{\alpha}\sum_{t=1}^{T} \sigma^2\eta_t^2 \|\nabla f_{1/\widehat{\rho}}(x_t)\|^2 \tag{55}$$

with probability at least $1 - \delta/2$ for any $\alpha \geq b\max_{t\in[T]} M_t = {}^{2b\widehat{\rho}\eta_1\sigma G}/_{(\widehat{\rho}-\rho)}$. Then, by combining the above inequality with (52), we obtain

$$\sum_{t=1}^{T} \eta_t \left(\frac{\widehat{\rho} - \rho}{\rho} - \frac{a\sigma^2\eta_t}{\alpha}\right) \|\nabla f_{1/\widehat{\rho}}(x_t)\|^2 \leq \Delta_1 + 2\alpha\log(4/\delta) + \widehat{\rho}\gamma^2\left(2\sigma^2(4e\theta)^{2\theta} \log^{2\theta}(4/\delta) + G^2\right)\log(eT) \tag{56}$$

with probabilty at least $1 - \delta$. Set $\widehat{\rho} = 3\rho$ and $\alpha = \eta_1 \max\{{}^{2b\widehat{\rho}\sigma G}/_{(\widehat{\rho}-\rho)}, 3a\sigma^2\}$ to make

$$\frac{\widehat{\rho} - \rho}{\widehat{\rho}} - \frac{a\sigma^2\eta_t}{\alpha} \geq \frac{2}{3} - \frac{\eta_t}{3\eta_1} \geq \frac{1}{3} \tag{57}$$

Then we can get

$$\sum_{t=1}^{T} \eta_t \|\nabla f_{1/\widehat{\rho}}(x_t)\|^2 \leq 3\Delta_1 + 6\alpha\log(4/\delta) + 9\rho\gamma^2\left(2\sigma^2(4e\theta)^{2\theta} \log^{2\theta}(4/\delta) + G^2\right)\log(eT) \tag{58}$$

and, depending on the specific value of $\theta$, $\alpha$ can be expressed as

$$\alpha = \begin{cases} 6\gamma\sigma^2, & \theta = 1/2, \\ 3\gamma \cdot \max\left\{(4\theta)^\theta\rho\sigma G, (4\theta)^{2\theta}e^2\sigma^2\right\}, & \theta \in (1/2, 1], \\ 3\gamma \cdot \max\left\{2\rho\sigma G\log^{\theta-1}(4T/\delta), c(\theta)\sigma^2\right\}, & \theta > 1. \end{cases} \tag{59}$$

where $c(\theta) := (2^{2\theta+1} + 2)\Gamma(2\theta + 1) + 2^{3\theta}\Gamma(3\theta + 1)/3$ and $\Gamma(\cdot)$ represents the Gamma funciton. By substituting $\alpha$ into (58) and multiplying both sides by $(T\eta_T)^{-1} = {}^1/_{\gamma\sqrt{T}}$, we obtain

$$\frac{1}{T}\sum_{t=1}^{T} \|\nabla f_{1/3\rho}(x_t)\|^2 \leq \frac{3\Delta_1}{\gamma\sqrt{T}} + \frac{9\rho\gamma\left(2\sigma^2(4e\theta)^{2\theta} \log^{2\theta}(4/\delta) + G^2\right)\log(eT)}{\sqrt{T}}$$

$$+ \frac{36\log(4/\delta)\left\{(1 + (4\theta)^{2\theta} + c(\theta))\sigma^2 + \mathbb{1}_{\theta\in(1/2,1]}(4\theta)^\theta\rho\sigma G + \mathbb{1}_{\theta>1}\rho\sigma G\log^{\theta-1}(4T/\delta)\right\}}{\sqrt{T}} \tag{60}$$

Ignoring certain constants, we can simplify the above inequality as

$$\frac{1}{T}\sum_{t=1}^{T} \|\nabla f_{1/3\rho}(x_t)\|^2 \lesssim \frac{\Delta_1 + \rho\log(T)\left(\log^{2\theta}(1/\delta)\sigma^2 + G^2\right)}{\sqrt{T}} + \frac{\mathbb{1}_{\theta>1/2}\rho\sigma G\log(1/\delta)\log^{\max\{\theta,1\}-1}(T/\delta)}{\sqrt{T}} \tag{61}$$

### D.6. Proof of Corollary 4.5

Upon fixing the time horizon $T$, setting $\eta_t$ to $\gamma/\sqrt{T}$ and requiring $\alpha = \gamma \max\{2b\widehat{\rho}\sigma G/(\widehat{\rho}-\rho),3a\sigma^2\}/\sqrt{T}$, then (60) reduces to

$$
\frac{1}{T}\sum_{t=1}^{T}\left\|\nabla f_{1/3\rho}(x_t)\right\|^2 \leq \frac{3\Delta_1}{\gamma\sqrt{T}} + \frac{9\rho\gamma\left(2\sigma^2(4e\theta)^{2\theta}\log^{2\theta}(4/\delta) + G^2\right)}{\sqrt{T}}
$$
$$
+ \frac{36\log(4/\delta)\left\{(1+(4\theta)^{2\theta}+c(\theta))\sigma^2 + \mathbb{1}_{\theta\in(1/2,1]}(4\theta)^\theta\rho\sigma G + \mathbb{1}_{\theta>1}\rho\sigma G\log^{\theta-1}(4T/\delta)\right\}}{T} \quad (62)
$$

Let $\gamma = \sqrt{\Delta_1/3\rho(2\sigma^2(4e\theta)^{2\theta}\log^{2\theta}(4/\delta)+G^2)}$, then we have

$$
\frac{1}{T}\sum_{t=1}^{T}\left\|\nabla f_{1/3\rho}(x_t)\right\|^2 \leq \frac{3\sqrt{3\Delta_1\rho\left(2\sigma^2(4e\theta)^{2\theta}\log^{2\theta}(4/\delta)+G^2\right)}}{\sqrt{T}}
$$
$$
+ \frac{36\log(4/\delta)\left\{(1+(4\theta)^{2\theta}+c(\theta))\sigma^2 + \mathbb{1}_{\theta\in(1/2,1]}(4\theta)^\theta\rho\sigma G + \mathbb{1}_{\theta>1}\rho\sigma G\log^{\theta-1}(4T/\delta)\right\}}{T}
$$
$$
\leq \frac{3\sqrt{3\Delta_1\rho}\left(\sqrt{2}(4e\theta)^\theta\log^\theta(4/\delta)\sigma+G\right)}{\sqrt{T}}
$$
$$
+ \frac{36\log(4/\delta)\left\{(1+(4\theta)^{2\theta}+c(\theta))\sigma^2 + \mathbb{1}_{\theta\in(1/2,1]}(4\theta)^\theta\rho\sigma G + \mathbb{1}_{\theta>1}\rho\sigma G\log^{\theta-1}(4T/\delta)\right\}}{T}
$$
$$
\lesssim \sqrt{\frac{\Delta_1\rho}{T}}\left(\log^\theta(1/\delta)\sigma + G\right) + \frac{\log(1/\delta)\sigma^2}{T} + \frac{\mathbb{1}_{\theta>1/2}\cdot\rho\sigma G\log^{\max\{\theta,1\}-1}(T/\delta)\log(1/\delta)}{T} \quad (63)
$$

### D.7. Proof of Thereom 5.1

*Proof.* We proceed analogously to the proof of Lemma 4.1, with the only changes being that we invoke the non-expansiveness of projection and substitute $g_t^{\mathrm{clip}}$ for $g_t$, that is,

$$
\begin{aligned}
f_{1/\widehat{\rho}}(x_{t+1}) &= f(\widehat{x}_{t+1}) + \frac{\widehat{\rho}}{2}\|\widehat{x}_{t+1} - x_{t+1}\|^2 \\
&\leq f(\widehat{x}_t) + \frac{\widehat{\rho}}{2}\|\widehat{x}_t - x_{t+1}\|^2 \\
&= f(\widehat{x}_t) + \frac{\widehat{\rho}}{2}\|\widehat{x}_t - \Pi_{\mathcal{X}}\left(x_t - \eta_t g_t^{\mathrm{clip}}\right)\|^2 \\
&= f(\widehat{x}_t) + \frac{\widehat{\rho}}{2}\|\Pi_{\mathcal{X}}(\widehat{x}_t) - \Pi_{\mathcal{X}}\left(x_t - \eta_t g_t^{\mathrm{clip}}\right)\|^2 \\
&\overset{(b)}{\leq} f(\widehat{x}_t) + \frac{\widehat{\rho}}{2}\|\widehat{x}_t - x_t + \eta_t g_t^{\mathrm{clip}}\|^2 \\
&= f(\widehat{x}_t) + \frac{\widehat{\rho}}{2}\|\widehat{x}_t - x_t\|^2 + \widehat{\rho}\eta_t\left\langle\widehat{x}_t - x_t, g_t^{\mathrm{clip}}\right\rangle + \frac{\widehat{\rho}\eta_t^2}{2}\|g_t^{\mathrm{clip}}\|^2 \\
&= f_{1/\widehat{\rho}}(x_t) + \widehat{\rho}\eta_t\langle\widehat{x}_t - x_t, \nabla_t\rangle + \widehat{\rho}\eta_t\langle\widehat{x}_t - x_t, \epsilon_t\rangle + \frac{\widehat{\rho}\eta_t^2}{2}\|\epsilon_t + \nabla_t\|^2 \\
&\leq f_{1/\widehat{\rho}}(x_t) + \widehat{\rho}\eta_t\left(f(\widehat{x}_t) - f(x_t) + \frac{\rho}{2}\|\widehat{x}_t - x_t\|^2\right) + \widehat{\rho}\eta_t\langle\widehat{x}_t - x_t, \epsilon_t\rangle + \widehat{\rho}\eta_t^2(\|\epsilon_t\|^2 + G^2) \\
&= f_{1/\widehat{\rho}}(x_t) - \widehat{\rho}\eta_t\left[\left(f(x_t) + \frac{\widehat{\rho}}{2}\|x_t - x_t\|^2\right) - \left(f(\widehat{x}_t) + \frac{\widehat{\rho}}{2}\|x_t - \widehat{x}_t\|^2\right) + \frac{\widehat{\rho}-\rho}{2}\|x_t - \widehat{x}_t\|^2\right] \\
&\quad + \widehat{\rho}\eta_t\langle\widehat{x}_t - x_t, \epsilon_t\rangle + \widehat{\rho}\eta_t^2(\|\epsilon_t\|^2 + G^2) \\
&\leq f_{1/\widehat{\rho}}(x_t) - \widehat{\rho}\eta_t(\widehat{\rho}-\rho)\|\widehat{x}_t - x_t\|^2 + \widehat{\rho}\eta_t\langle\widehat{x}_t - x_t, \epsilon_t\rangle + \widehat{\rho}\eta_t^2(\|\epsilon_t\|^2 + G^2) \\
&= f_{1/\widehat{\rho}}(x_t) - \frac{\widehat{\rho}-\rho}{\widehat{\rho}}\eta_t\|\nabla f_{1/\widehat{\rho}}(x_t)\|^2 + \widehat{\rho}\eta_t\langle\widehat{x}_t - x_t, \epsilon_t\rangle + \widehat{\rho}\eta_t^2(\|\epsilon_t\|^2 + G^2), \quad (64)
\end{aligned}
$$

where $\epsilon_t$ is redefined as the clipped error, that is, $\epsilon_t := g_t^{\mathrm{clip}} - \nabla_t$. Then, by rearranging the terms, we obtain

$$\frac{\widehat{\rho} - \rho}{\widehat{\rho}} \eta_t \left\| \nabla f_{1/\widehat{\rho}}(x_t) \right\|^2 \le f_{1/\widehat{\rho}}(x_t) - f_{1/\widehat{\rho}}(x_{t+1}) + \widehat{\rho}\eta_t \langle \epsilon_t, \widehat{x}_t - x_t \rangle + \widehat{\rho}\eta_t^2(G^2 + \|\epsilon_t\|^2), \tag{65}$$

Next, we split $\epsilon_t$ into $\epsilon_t^u + \epsilon_t^b$, where $\epsilon_t^u = g_t^{\mathrm{clip}} - \mathbb{E}_t[g_t^{\mathrm{clip}}]$ and $\epsilon_t^b = \mathbb{E}_t[g_t^{\mathrm{clip}}] - \nabla_t$. With this decomposition, the left-hand side of (65) can be rewritten as

$$\frac{\widehat{\rho} - \rho}{\widehat{\rho}} \eta_t \left\| \nabla f_{1/\widehat{\rho}}(x_t) \right\|^2 \le f_{1/\widehat{\rho}}(x_t) - f_{1/\widehat{\rho}}(x_{t+1}) + \widehat{\rho}\eta_t \langle \epsilon_t^u, \widehat{x}_t - x_t \rangle + 2\widehat{\rho}\eta_t^2(\|\epsilon_t^b\|^2 - \mathbb{E}[\|\epsilon_t^b\|^2])$$
$$+ \widehat{\rho}\eta_t^2(G^2 + 2\|\epsilon_t^u\|^2 + 2\mathbb{E}[\|\epsilon_t^b\|^2]) + \widehat{\rho}\eta_t \langle \epsilon_t^b, \widehat{x}_t - x_t \rangle \tag{66}$$

Summing both sides from $t = 1$ to $T$, we obtain

$$\sum_{t=1}^T \frac{\widehat{\rho} - \rho}{\widehat{\rho}} \eta_t \left\| \nabla f_{1/\widehat{\rho}}(x_t) \right\|^2 \le \Delta_1 + \underbrace{\sum_{t=1}^T \widehat{\rho}\eta_t \langle \epsilon_t^u, \widehat{x}_t - x_t \rangle}_{①} + \underbrace{\sum_{t=1}^T 2\widehat{\rho}\eta_t^2(\|\epsilon_t^u\|^2 - \mathbb{E}_t[\|\epsilon_t^u\|^2])}_{②}$$
$$+ \underbrace{\sum_{t=1}^T \left\{ \widehat{\rho}\eta_t^2(G^2 + 2\|\epsilon_t^b\|^2 + 2\mathbb{E}_t[\|\epsilon_t^u\|^2]) + \widehat{\rho}\eta_t \langle \epsilon_t^b, \widehat{x}_t - x_t \rangle \right\}}_{③} \tag{67}$$

Observe that, with the specific choices $\lambda_t = \max\{2G, \sigma t^{1/p}\}$ and $\eta_t = \min\{1/\lambda_t, 1/(G\sqrt{t})\}$, it follows that

$$\eta_t \lambda_t \le 1; \quad (\sigma/\lambda_t)^p \le 1/t; \quad G^2 \eta_t^2 \le 1/t \tag{68}$$

In the analysis that follows, we choose $\widehat{\rho} = 2\rho$. In addition, we will use the following lemma to facilitate our analysis.

**Lemma D.1.** *(Lemma 10 from (Liu & Zhou, 2023)) Let $\mathcal{F}_t := \sigma(g_1, \ldots, g_t)$ be the filtration generated by the past history of the stochastic gradients in Algorithm 2 and $\mathbb{E}_t[\cdot] := \mathbb{E}[\cdot|\mathcal{F}_{t-1}]$. Define $\epsilon_t = g_t^{\mathrm{clip}} - \mathbb{E}_t[g_t], \epsilon_t^u = g_t^{\mathrm{clip}} - \mathbb{E}_t[g_t^{\mathrm{clip}}]$ and $\epsilon_t^b = \mathbb{E}_t[g_t^{\mathrm{clip}}] - \mathbb{E}_t[g_t]$. Note that $\epsilon_t$ and $\epsilon^u$ are $\mathcal{F}_t$-measurable but $\epsilon_t^b$ is $\mathcal{F}_{t-1}$-measurable. Suppose that Assumption 1.1 and 3.2 hold. If $\lambda_t \ge 2G$, then we can obtain the following inequalities hold almost surely.*

$$\mathbb{E}_t\left[\|\epsilon_t\|^2\right], \mathbb{E}_t\left[\|\epsilon_t^u\|^2\right], \|\epsilon_t^b\|^2 \le 10\sigma^p \lambda_t^{2-p};$$
$$\|\epsilon_t^u\| \le 2\lambda_t, \|\epsilon_t^b\| \le 2\sigma^p \lambda_t^{1-p}$$

**High-probability upper bound of ①.**

We first notice that

$$|\widehat{\rho}\eta_t \langle \widehat{x}_t - x_t, \epsilon_t^u \rangle| \le \widehat{\rho}\eta_t \|\widehat{x}_t - x_t\| \|\epsilon_t^u\| \overset{\text{Lemmas C.3, D.1}}{\le} \widehat{\rho}\eta_t \cdot \frac{2G}{\widehat{\rho} - \rho} \cdot 2\lambda_t = 8G \cdot \lambda_t \eta_t \overset{(68)}{\le} 8G \tag{69}$$

we also have

$$\mathbb{E}_t[|\widehat{\rho}\eta_t \langle \widehat{x}_t - x_t, \epsilon_t^u \rangle|^2] \le 4\rho^2 \eta_t^2 \|\widehat{x}_t - x_t\|^2 \mathbb{E}_t[\|\epsilon_t^u\|^2] \overset{\text{Lemmas C.3, D.1}}{\le} 160G^2(\eta_t \lambda_t)^2 \cdot (\sigma/\lambda_t)^p \overset{(68)}{\le} \frac{160G^2}{t}$$

Hence, we can get

$$\sum_{t=1}^T \mathbb{E}_t[|\widehat{\rho}\eta_t \langle \widehat{x}_t - x_t, \epsilon_t^u \rangle|^2] \le 160G^2 \log(eT) \tag{70}$$

Note that $\mathbb{E}_t[\widehat{\rho}\eta_t\langle\widehat{x}_t - x_t, \epsilon_t^u\rangle] = 0$. To apply Lemma C.7, we set $R = 8G$ and $F = 160G^2\log(eT)$, then with the probability at least $1 - \delta/2$, we have

$$
\begin{aligned}
\sum_{t=1}^{T}|\widehat{\rho}\eta_t\langle\widehat{x}_t - x_t, \epsilon_t^u\rangle| &\leq \frac{2R}{3}\log(4/\delta) + \sqrt{2F\log(4/\delta)} \\
&\leq \frac{16G\log(4/\delta)}{3} + \sqrt{320G^2\log(eT)\log(4/\delta)} \\
&\leq 24G\left(\log(4/\delta) + \sqrt{\log(eT)\log(4/\delta)}\right)
\end{aligned}
\tag{71}
$$

**High-probability upper bound of ②**

Firstly, we have

$$
\left|\widehat{\rho}\eta_t^2(\|\epsilon_t^u\|^2 - \mathbb{E}_t\|\epsilon_t^u\|^2)\right| \leq \widehat{\rho}\eta_t^2(\|\epsilon_t^u\|^2 + \mathbb{E}_t\|\epsilon_t^u\|^2) \overset{\text{Lemma D.1}}{\leq} 8\widehat{\rho}(\lambda_t\eta_t)^2 \overset{(68)}{\leq} 16\rho
\tag{72}
$$

then we get

$$
\begin{aligned}
\mathbb{E}_t\left[\left|\widehat{\rho}\eta_t^2(\|\epsilon_t^u\|^2 - \mathbb{E}_t\|\epsilon_t^u\|^2)\right|^2\right] &= \mathbb{E}_t\left[\widehat{\rho}^2\eta_t^4\left(\|\epsilon_t^u\|^4 - 2\langle\|\epsilon_t^u\|^2, \mathbb{E}_t\|\epsilon_t^u\|^2\rangle + (\mathbb{E}_t\|\epsilon_t^u\|^2)^2\right)\right] \\
&= \widehat{\rho}^2\eta_t^4\left(\mathbb{E}_t\|\epsilon_t^u\|^4 - 2(\mathbb{E}_t\|\epsilon_t^u\|^2)^2 + (\mathbb{E}_t\|\epsilon_t^u\|^2)^2\right) \\
&\leq \widehat{\rho}^2\eta_t^4\mathbb{E}_t\|\epsilon_t^u\|^4 \overset{\|\epsilon_t^u\|^2\leq 4\lambda_t^2}{\leq} \widehat{\rho}^2\eta_t^4\cdot 4\lambda_t^2\mathbb{E}_t\|\epsilon_t^u\|^2 \overset{\text{Lemma D.1}}{\leq} 4\widehat{\rho}^2(\eta_t\lambda_t)^4\cdot 10(\sigma/\lambda_t)^p \\
&\overset{(68)}{\leq} \frac{160\rho^2}{t}
\end{aligned}
\tag{73}
$$

Now, we have the upper bound

$$
\sum_{t=1}^{T}\mathbb{E}_t\left[\left|\widehat{\rho}\eta_t^2(\|\epsilon_t^u\|^2 - \mathbb{E}_t\|\epsilon_t^u\|^2)\right|^2\right] \leq 160\rho^2\log(eT)
\tag{74}
$$

Similarly, we set $R = 16\rho$ and $F = 160\rho^2\log(eT)$, then apply Lemma C.7 and get the following inequality holds with the probability at least $1 - \delta/2$

$$
\begin{aligned}
\sum_{t=1}^{T}\left|\widehat{\rho}\eta_t^2(\|\epsilon_t^u\|^2 - \mathbb{E}_t\|\epsilon_t\|^2)\right| &\leq \frac{2R\log(4/\delta)}{3} + \sqrt{2F\log(4/\delta)} \\
&\leq \frac{32}{3}\rho\log(4/\delta) + \rho\sqrt{320\log(eT)\log(4/\delta)} \\
&\leq 24\rho\left(\log(4/\delta) + \sqrt{\log(eT)\log(4/\delta)}\right)
\end{aligned}
\tag{75}
$$

**Almost sure upper bound of ③**

The following inequality holds almost surely

$$
\begin{aligned}
③ &\leq \sum_{t=1}^{T}2\rho\left\{\eta_t^2 G^2 + 2\eta_t^2\left\|\epsilon_t^b\right\|^2 + 2\eta_t^2\mathbb{E}_t[\|\epsilon_t^u\|^2]) + \eta_t\|\epsilon_t^b\|\|\widehat{x}_t - x_t\|\right\} \\
&\overset{\text{Lemma D.1}}{\leq} \sum_{t=1}^{T}2\rho\left\{\eta_t^2 G^2 + 40(\sigma/\lambda_t)^p(\eta_t\lambda_t)^2 + 4\rho^{-1}G(\sigma/\lambda_t)^p(\lambda_t\eta_t)\right\} \\
&\overset{(68)}{\leq} 2\rho\log(eT) + 80\rho\log(eT) + 8G\log(eT) \leq 82(\rho + G)\log(eT)
\end{aligned}
\tag{76}
$$

Finally, with the probability at least $1 - \delta$, we get

$$
\begin{aligned}
\frac{1}{T}\sum_{t=1}^{T}\left\|\nabla f_{1/2\rho}(x_t)\right\|^2 &\leq \frac{2(\Delta_1 + ① + ② + ③)}{T\eta_T} \\
&\leq 2\left\{\Delta_1 + 82\left(\rho + G\right)\left(\log(eT) + \log(4/\delta) + \sqrt{\log(eT)\log(4/\delta)}\right)\right\} \cdot \max\left\{\frac{G}{\sqrt{T}}, \frac{\sigma}{T^{\frac{p-1}{p}}}\right\} \\
&\leq 2\left\{\Delta_1 + 82\left(\rho + G\right)\left(\frac{3}{2}\log(eT) + \frac{3}{2}\log(4/\delta)\right)\right\} \cdot \max\left\{\frac{G}{\sqrt{T}}, \frac{\sigma}{T^{\frac{p-1}{p}}}\right\} \\
&\leq 246\left\{\Delta_1 + \left(\rho + G\right)\log(4eT/\delta)\right\} \cdot \max\left\{\frac{G}{\sqrt{T}}, \frac{\sigma}{T^{\frac{p-1}{p}}}\right\}
\end{aligned}
\tag{77}
$$

If we fix the time-horizon $T$ and set $\lambda_t \equiv \max\{2G, \sigma T^{1/p}\}$ and $\eta_t \equiv \min\{1/\lambda_t, 1/(G\sqrt{T})\}$, then (68) becomes

$$
\eta_t\lambda_t \leq 1; \quad (\sigma/\lambda_t)^p \leq 1/T; \quad G^2\eta_t^2 \leq 1/T
\tag{78}
$$

Note that $|\widehat{\rho}\eta_t\langle\widehat{x}_t - x_t, \epsilon_t^u\rangle| \leq 8G$ and $\mathbb{E}_t[|\widehat{\rho}\eta_t\langle\widehat{x}_t - x_t, \epsilon_t^u\rangle|^2] \leq 160G^2/T$. We set $R = 8G$ and $F = \sum_{t=1}^{T}\mathbb{E}_t[|\widehat{\rho}\eta_t\langle\widehat{x}_t - x_t, \epsilon_t^u\rangle|^2] \leq 160G^2$, then with the probability at least $1 - \delta/2$, we have

$$
① \leq 24G\left(\log(4/\delta) + \sqrt{\log(4/\delta)}\right)
\tag{79}
$$

Similarly, since $\left|\widehat{\rho}\eta_t^2(\|\epsilon_t^u\|^2 - \mathbb{E}_t\|\epsilon_t^u\|^2)\right| \leq 16\rho$ and $\mathbb{E}_t\left[\left|\widehat{\rho}\eta_t^2(\|\epsilon_t^u\|^2 - \mathbb{E}_t\|\epsilon_t^u\|^2)\right|^2\right] \leq 160\rho^2$ we set $R = 16\rho$ and $F = \sum_{t=1}^{T}\mathbb{E}_t\left[\left|\widehat{\rho}\eta_t^2(\|\epsilon_t^u\|^2 - \mathbb{E}_t\|\epsilon_t^u\|^2)\right|^2\right] \leq 160\rho^2$. With probability $1 - \delta$, we have

$$
② \leq 24\rho\left(\log(4/\delta) + \sqrt{\log(4/\delta)}\right)
\tag{80}
$$

The upper bound of ③ becomes

$$
③ \leq 82(\rho + G)
\tag{81}
$$

With probability at least $1 - \delta$, we get

$$
\begin{aligned}
\frac{1}{T}\sum_{t=1}^{T}\left\|\nabla f_{1/2\rho}(x_t)\right\|^2 &\leq 2\left\{\Delta_1 + 82\left(\rho + G\right)\left(\log(4/\delta) + \sqrt{\log(4/\delta)} + 1\right)\right\} \cdot \max\left\{\frac{G}{\sqrt{T}}, \frac{\sigma}{T^{\frac{p-1}{p}}}\right\} \\
&\leq 2\left\{\Delta_1 + 82\left(\rho + G\right)\left(\frac{3}{2}\log(4e/\delta)\right)\right\} \cdot \max\left\{\frac{G}{\sqrt{T}}, \frac{\sigma}{T^{\frac{p-1}{p}}}\right\} \\
&\leq 246\left\{\Delta_1 + \left(\rho + G\right)\log(4e/\delta)\right\} \cdot \max\left\{\frac{G}{\sqrt{T}}, \frac{\sigma}{T^{\frac{p-1}{p}}}\right\}
\end{aligned}
\tag{82}
$$

$\square$

## E. In-Expectation Convergence of clip-SGD under the $p$-BCM noise

**Theorem E.1.** *Suppose that Assumptions 1.1 ,3.1 and 3.2 hold. Let $\lambda_t = 2G \vee \sigma t^{1/p}$ and $\eta_t = 1/\lambda_t \wedge 1/G\sqrt{t}$ in clip-SGD. Then $1/T\left\|\nabla f_{1/2\rho}(x_t)\right\|^2$ is*

$$
\mathcal{O}\left(\left(\Delta_1 + (G + \rho)\log T\right)\left(\frac{G}{\sqrt{T}} \vee \frac{\sigma}{T^{1-\frac{1}{p}}}\right)\right)
\tag{83}
$$

*Moreover, let $\lambda_t \equiv 2G \vee \sigma T^{1/p}$ and $\eta_t \equiv 1/\lambda_t \wedge 1/G\sqrt{T}$ after fixing $T$. Then we have*

$$
\mathcal{O}\left(\left(\Delta_1 + (G + \rho)\right)\left(\frac{G}{\sqrt{T}} \vee \frac{\sigma}{T^{1-\frac{1}{p}}}\right)\right)
\tag{84}
$$

Compared with Theorem 5.1, the only difference in Theorem E.1 is the disappearance of the $\log(1/\delta)$ term.

*Proof.* By setting $\widehat{\rho} = 2\rho$ and taking the condtional expectation $\mathbb{E}_t[\cdot]$ on both sides of (67), we obtain

$$
\sum_{t=1}^{T} \eta_t \mathbb{E}_t \left[ \left\| \nabla f_{1/2\rho}(x_t) \right\|^2 \right] \leq 2\Delta_1 + 2 \sum_{t=1}^{T} \left\{ 2\rho \eta_t^2 \left( G^2 + 2\mathbb{E}_t[\|\epsilon_t^b\|^2] + 2\mathbb{E}_t[\|\epsilon_t^u\|^2] \right) + 2\rho \eta_t \left\langle \epsilon_t^b, \widehat{x}_t - x_t \right\rangle \right\}
$$

$$
\leq 2\Delta_1 + 2 \sum_{t=1}^{T} \left\{ 2\rho \left( 1/t + 40(\sigma/\lambda_t)^p (\eta_t \lambda_t)^2 \right) + 2\rho \eta_t \| \epsilon_t^b \| \| \widehat{x}_t - x_t \| \right\}
$$

$$
\leq 2\Delta_1 + 2 \left\{ 2\rho \left( \log(eT) + 40 \log(eT) \right) \right\} + 16G \log(eT)
$$

$$
\leq 2\Delta_1 + 164(\rho + G) \log(eT) \leq 164 \left\{ \Delta_1 + (\rho + G) \log(eT) \right\} \tag{85}
$$

Taking full expectation and multiply both sides by $(T\eta_T)^{-1}$, we finally have

$$
\frac{1}{T} \mathbb{E} \left[ \left\| \nabla f_{1/2\rho}(x_t) \right\|^2 \right] \leq 164 \left\{ \Delta_1 + (\rho + G) \log(eT) \right\} \cdot \max \left\{ \frac{G}{\sqrt{T}}, \frac{\sigma}{T^{\frac{p-1}{p}}} \right\} \tag{86}
$$

If we fix the time-horizon $T$ and set $\lambda_t \equiv \max\{2G, \sigma T^{1/p}\}$ and $\eta_t \equiv \min\{1/\lambda_t, 1/(G\sqrt{T})\}$, then we have

$$
\frac{1}{T} \mathbb{E} \left[ \left\| \nabla f_{1/2\rho}(x_t) \right\|^2 \right] \leq 164 \left\{ \Delta_1 + (\rho + G) \right\} \cdot \max \left\{ \frac{G}{\sqrt{T}}, \frac{\sigma}{T^{\frac{p-1}{p}}} \right\} \tag{87}
$$

$\square$

