# OpenReview forum: "Stochastic Gradient Methods under Heavy-Tailed Noises in Weakly Convex Optimization"
_ICML.cc/2026/Conference — ICML 2026 regular_

### Official Review · Reviewer_YzVg · 2026-03-02

**Soundness:** 2
**Presentation:** 2
**Significance:** 1
**Originality:** 3
**Overall Recommendation:** 3
**Confidence:** 4

**Summary:**

This work mainly focuses on stochastic gradient methods under heavy-tailed noises in weakly convex optimization. Particularly, they analyzed the convergence behavior of vanilla SGD and Clip-SGD. Numerical experiments validate the authors’ theoretical findings.

**Compliance With Llm Reviewing Policy:**

Affirmed.

**Key Questions For Authors:**

1. The contribution of this work is limited. The important reference is missed by the authors; see below.
“Stochastic Weakly Convex Optimization Under Heavy-Tailed Noises, 2025.”
I find that similar cases have been discussed in the above reference. It seems the theoretical findings of this work are just an extension of the missed references. In addition, the key reference “Tight Lower Bounds and Optimal Algorithms for Stochastic Nonconvex Optimization with Heavy-Tailed Noise” is also missing.


2. The experiments do not strongly support the theoretical findings of the authors’ claim. As the authors emphasized that many applications in large language models, image classification, etc., satisfied the assumption pointed out by the authors. To better support the theoretical findings, experiments on complex models should be provided.

3. Several typos appear in the article, see line 94 in page 2, “for vanilla SGDin”; line 30 in page 1, $ \epsilon(x, \xi)=g(x, \xi)-g(x)$ should be $ \epsilon(x, \xi)=g(x, \xi)-f(x)$.

**Limitations:**

See weakness

**Strengths And Weaknesses:**

Strengths
The authors discussed several classical stochastic optimization algorithms, such as vanilla SGD and Clip-SGD. More specifically, they showed the case of vanilla SGD under the case of $p$-th the central moment assumption, the stochastic gradient noise is sub-Weibull. Addition, the authors established a high-probability complexity bound for Clip-SGD, which exhibited only a polylogarithmic dependence on $1/\delta$.

Weaknesses
1. The contribution of this work is limited. The important reference is missed by the authors; see below.
“Stochastic Weakly Convex Optimization Under Heavy-Tailed Noises, 2025.”
I find that similar cases have been discussed in the above reference. It seems the theoretical findings of this work are just an extension of the missed references. In addition, the key reference “Tight Lower Bounds and Optimal Algorithms for Stochastic Nonconvex Optimization with Heavy-Tailed Noise” is also missing.


2. The experiments do not strongly support the theoretical findings of the authors’ claim. As the authors emphasized that many applications in large language models, image classification, etc., satisfied the assumption pointed out by the authors. To better support the theoretical findings, experiments on complex models should be provided.

3. Several typos appear in the article, see line 94 in page 2, “for vanilla SGDin”; line 30 in page 1, $ \epsilon(x, \xi)=g(x, \xi)-g(x)$ should be $ \epsilon(x, \xi)=g(x, \xi)-f(x)$.

---

> ### Author Rebuttal · Authors · 2026-03-29
>
> Our response to your constructive comments on this work is as follows:
>
> **Q1**: As for the article you actively mentioned that is closely related to our submission but not cited, it is the early preprint of the submission. For the theoretical results in this paper, we declare that they are all unpublished and completely new. Regarding the literature 'Tight Lower Bounds and Optimal Algorithms for Stochastic Non Convex Optimization with Heavy Tailed Noise', they actually study the lower complexity of accelerated methods (such as Normlized-SGD with variance reduction (with or without gradient clipping) , some second-order methods), while can't be used to make the comparsion in our setting. We will discuss this literature in the revision.
>
> **Q2**: We understand your concerns about not evaluating on real data and models. In fact, the reason we use explicit weakly convex objective, synthetic data, and inject noise is to be able to determine the class of the objective and noise distribution parameters to strengthen the validation of theoretical insights. This is impossible to achieve in actual training, especially when training language models. Thus, due to the inability to determine the  class of the objectives and the heavy-tailed distribution, all hyperparameters of the methods can only be tuned empirically, as in previous work. Nevertheless, we are still willing to include some experiments on fine-tuning the language model in revision.
>
> **Q3**：Thank you for pointing out the typo. We will continue to polish our submission.

---

### Official Review · Reviewer_2jG7 · 2026-03-08

**Soundness:** 3
**Presentation:** 3
**Significance:** 3
**Originality:** 3
**Overall Recommendation:** 3
**Confidence:** 2

**Summary:**

This paper studies SGD methods under heavy-tailed gradient noise for weakly convex optimization. First, the paper establishes convergence guarantees without gradient clipping, normalization under two different heavy-tailed assumptions, the bounded p-th central moment assumption and the sub-Weibull assumption. Next, for Clip-SGD, a high-probability complexity bound is obtained in the weakly convex setting.
If the noise variance is bounded, the results in this paper on sample complexity reduces to well known results for gradient-based methods for weakly convex optimization in the literature. Finally, some numerical experiments are provided to illustrate the theory.

**Compliance With Llm Reviewing Policy:**

Affirmed.

**Key Questions For Authors:**

(1) On page 4, there should be some space before (Zhang et al., 2020... The same issue can be said about (Davis et al. 2018) on page 1,
(Ahn et al., 2024) on page 1 etc.

(2) In the paragraph after equation (3), you have $\mathcal{N}_{\mathcal{X}}(x):=\{d:\langle d,y-x\rangle\leq 0,\forall y\in\mathcal{X}\}$.
It is better to replace $d$ by some other letter so that it is not confused with dimension $d$.

(3) On page 1, when you talk about bounded variance and sub-Gaussian assumption, please mention it is uniformly in $x$ because later in Assumption 1.1. ($p$-BCM), you made that explicit.

(4) In the last paragraph on page 2, SGDin should be SGD in.

(5) On page 7, The dependence on faliure probability should be The dependence on failure probability.

(6) In Table 1, for FEMA, Bouded should be Bounded.

(7) In the statement of Lemma C.4., what is $M_{t}$?

(8) In the first sentence of the proof of Lemma 4.1., Mreau envelope is a typo.

**Limitations:**

Yes.

**Strengths And Weaknesses:**

*Strengths

(1) There is an extensive literature review. There are adequate discussions on the assumptions and results by comparing to the literature. Table 1 nicely summarizes the results obtained in this paper and all the previous existing results in the literature to make the comparison easy to read.

(2) The analysis I believe is rigorous. The detailed technical lemmas and proofs are provided to support the main theoretical results in the paper.

*Weaknesses

(1) The main theoretical results are obtained for SGD either in a compact domain or Clip-SGD. Personally, I think heavy-tailed gradient noise is most interesting for unclipped, unnormalized SGD in unbounded domain. The reason is that heavy tails only exist in unbounded domain. When you restrict to a compact domain, the effect of heavy tails only occur when the algorithm exits the domain, but then it gets immediately projected back. For clipped SGD or normalized SGD, some sort of boundedness is created, and that also mitigates the effect from heavy tails. But I understand that they are popular and often studied in the literature. But it would be nice if there are some unclipped, unnormalized SGD results in unconstrained optimization, which will make the paper stronger.

(2) One limitation as the author(s) pointed out is that assuming $p$-BCM, the stepsizes of SGD and Clip-SGD depend on the typically unknown value of $p$ to achieve improved convergence rate. But that can be left as a future research direction.

---

> ### Author Rebuttal · Authors · 2026-03-27
>
> Before responding to your constructive comments, we would like to highlight two strengths of our work that you did not mention.
>
> - **First tight complexity upper bound of vanilla SGD  under $p$-BCM noise for weakly convex optimization**. **[R1, Theorem 5.5]** establishes a lower bound for vanilla SGD under $p$-BCM noise in Hölder-smooth minimization: specifically, there exists a $(0,L_\nu,\nu)$-smooth and $G$-Lipschitz function $f$ such that, for sufficiently small $\varepsilon$, if the domain diameter satisfies $D \geq \Delta_1/\varepsilon$ (or unbonded), then the sample complexity of vanilla SGD with $\eta = t^{-r}$ to achieve $E[|\nabla f(x)|] \leq \varepsilon$ is at least $\Omega(\varepsilon^{-\frac{2}{1-r}} + \varepsilon^{-\frac{p+\nu-2}{r(p-1)(\nu-1)}})$. When $\nu = 2$ and $r=1/p$, this reduces to the standard smooth case (a special case of weakly convex optimization) $\Omega (\varepsilon^{-\frac{2p}{p-1}})$, which matches the upper bound in Theorem 4.2 with.
>
> - **Refined weakly convex analysis is better.** **[R1]** highlights that it remains unclear whether vanilla SGD with $O(1)$ mini-batch can achieve in-expectation convergence for standard smooth objectives (see discussion in **[Pag. 22, R1]**). They also employ a Moreau envelope based analysis to obtain a sample complexity of $O(\varepsilon^{-\frac{2p}{p-1}})$ for achieving $E[|\nabla f(x)|^p]\leq \varepsilon^p$, but this requires a batch size of $O(\varepsilon^{-\frac{2}{p(p-1)}})$ (see **[Theorem 5.10, R1]**). Our analysis provides a convergnfence guarantee of vanillia SGD with 1 batch-size for a broader class of functions, i.e., weakly convex functions.
>
>   > [R1] Fatkhullin, et al. https://arxiv.org/abs/2508.04860v1
>
> ---
>
> **For the key questions:**  We thank you for pointing out the typos and for your helpful writing suggestions. Regarding $M_t$ mentioned in (7), it is in fact the upper bound for $\sigma_{t-1}$. In Lemma C.4, we state that "*where $\sigma_t$ is …… and bounded by $M_{t+1}$*" (see line 792 in the submission).
>
> ---
>
> **For the Weakness:**
>
> **(1).** First, we point out that **vanilla SGD is hard to guarantee the convergence under heavy-tailed gradient noise in unconstrained optimization even for quadratic functions.**  For example, let the constraint domain be $R$, and $f=E_\xi[(1/2)x^2+\xi]$, where $\xi$ is  zero-mean noise with infinite variance and bounded $p$-th moment. After one step of SGD starting from $x_1 = 0$, we have $E[f(x_2)]=1/2E[|f^\prime (x_2)|^2]=1/2E[|\eta_1\xi|^2]= +\infty$. Moreover, we think that restricting the domain from unbounded to bounded does not eliminate the heavy-tailed nature of the noise. In our numerical experiments, we empirically observe that SGD is much less likely to achieve high-probability convergence compared to Clip-SGD. Technically, although the martingale term $\langle g_t- \nabla f(x_t),x_t-\hat{x}\_t \rangle $ in Lemma 4.1 has zero expectation, under heavy-tailed $p$-BCM noise it can still have infinite variance even on a bounded domain; hence, no existing concentration inequality can control the growth of $\sum_{t=1} |\langle g_t-\nabla f(x_t),x_t-\hat{x}_t\rangle|$ with high probability, indicating that the accumulation rate of $|\langle g_t-\nabla f(x_t),x_t-\hat{x}_t\rangle|$ is, with high probability, faster than the decay of the step size. For other unclipped and unnormalized SGD variants under heavy-tailed noise, we note that even for classical adaptive methods such as AdaGrad-Norm, **[R2]** shows that a bounded domain is required to guarantee convergence in expectation for convex functions. Nevertheless, it remains an interesting research direction to develop methods that can handle heavy-tailed noise and unconstrained domains without relying on clipping or normalization.
>
>    > [R2] Zijian. https://arxiv.org/abs/2508.07473v3
>
> **(2).** Under $p$-BCM noise, the upper bound complexity of SGD we provide with arbitrarily tuned $\eta$ is looser than the bound obtained with $p$-dependent $\eta$. This limitation also appears in existing work on Normalized-SGD (NSGD) in the smooth setting. At present, we do not have an answer as to whether the rate $O(\varepsilon^{-\frac{2p}{p-1}})$ is still achievable when $p$ is unknown, and we leave this as an interesting direction for future research.

---

> > ### Author Rebuttal · Reviewer_2jG7 · 2026-04-05
> >
> > The rebuttal is informative. I appreciate the author(s) pointing out two points of strengths that I did not mention in my report. The response to the weaknesses does not seem to resolve the concerns.

---

> > > ### Author Response · Authors · 2026-04-06
> > >
> > > Thanks for your rebuttal acknowledgement. We may not have explained clearly the two issues stemming from "Weaknesses":
> > > 1. Is convergence obvious for heavy-tailed noise within a bounded domain?
> > > 2. For unconstrained heavy-tailed optimization, is gradient clipping or normalization necessary?
> > >
> > > **For 1:**  For your statement in the "Weakness": *The reason is that heavy tails only exist in unbounded domain.*  We point out that the stochastic gradient $\nabla f(x_t,\xi)$ is only rely on the random variable when $x_t$ is already known. A bounded region does not statistically change the distribution of gradient noise because the heavy-tailed property of $\xi$ has not disappeared. Moreover, we have following
> > >  - From an intuitive perspective，even within bounded regions, we need to consider whether iteration points are always projected onto the boundary, since iteration points are more likely to go outside the region in the case of heavy tails. If this conjecture holds true, then the algorithm will still diverge if the optimal solution lies in the interior of the region.
> > >  - From a theoretical perspective, Once we utilize the nonexpansiveness of projection at the very beginning (as in the analysis of most previous works), the square norm of the stochastic gradient will still appear in the final descent lemma, which is uncontrollable under the $p$-BCM noise assumption, making it impossible to determine whether convergence has occurred. Therefore, we must adjust the classical analysis to accommodate the $p$-BCM noise assumption.
> > > - Over all, the first part of this work reveals that we should shift our focus from heavy-tailed gradients to constrained regions.  For example, for those interested algorithms that don't require gradient clipping and normalization, once we can prove that the algorithm is stable—that is, its sequence of iterations is always recursively restricted to a bounded region—even if the constrained region is unbounded, we can immediately prove that it converges in heavy-tailed environments. We have established key technologies to implement this approach for the future
> > >
> > > **For 2:** First, we point out that gradient clipping and normalization were not initially intended to serve convergence theory and are practically necessary. In fact, these techniques have long been popular in the deep learning community, particularly for stabilizing the training of large language models. **For example, [R1] emphasizes that using Clip-Adam instead of Adam yields better results when fine-tuning BERT. More recently, the Muon optimizer [R2] has gained popularity due to its widespread outperformance of Adam when training language models. It can be seen as an extension of Normalized-SGD when optimizing matrix variables.** Ensuring convergence in heavy-tailed environments improves the interpretability of these techniques in practice. Importantly, these models are unlikely to guarantee smoothness or convexity in practice. Nevertheless, from a purely theoretical research perspective, it is still meaningful to provide an algorithm that does not use gradient normalization and clipping and can converge in unconstrained heavy-tailed environments.
> > >
> > > > [R1]Mosbach, et al. "On the Stability of Fine-tuning BERT......". https://arxiv.org/abs/2006.04884
> > >
> > > > [R2] Jordan et al. "Muon: An optimizer for hidden layers in neural networks". https://kellerjordan.github.io/posts/muon/

---

### Official Review · Reviewer_nU1G · 2026-03-13

**Soundness:** 2
**Presentation:** 2
**Significance:** 2
**Originality:** 2
**Overall Recommendation:** 4
**Confidence:** 3

**Summary:**

The paper studies in-expectation and high-probability convergence guarantees for projected subgradient method, or its clipped version, under a constrained weakly convex setting. On the in-expectation side, the authors establish convergence of the projected subgradient descent under a bounded $p$-th noise moment (heavy-tailed). On the high-probability side, the authors establish convergence of vanilla subgradient descent under a sub-Weibull noise distribution, and of its clipped version under a heavy-tailed noise distribution.

**Compliance With Llm Reviewing Policy:**

Affirmed.

**Final Justification:**

The rebuttal has changed my evaluation and I believe there is some merits to accepting the paper. While I maintain that the topic of heavy-tailed noise is well-explored and that weakly convex functions is a niche topic, I am increasing my score and recommending weak accept.

**Key Questions For Authors:**

Overall, the work is interesting, however, due to the outlined weaknesses, I feel that it falls short of ICML standard. I am outlining some questions below and, depending on the authors' responses, I might be willing to increase the score.

1. Can it be verified that the bounds in Theorem 4.2 and 5.1 are tight for weakly convex settings? Or is it just a case of loose bounds?

2. How do the results in this work compare to adaptive methods under similar assumptions (e.g., heavy-tailed noise)?

**Limitations:**

Yes

**Strengths And Weaknesses:**

### Strengths

* A broad range of results is established under various noise settings, with some of them not known before, e.g., in-expectation
convergence of projected vanilla subgradient descent under heavy-tail noise and compact constraint set.

* The convergence results of Theorem 42.2 under a decaying learning rates can match the standard complexity results under a bounded variance condition. The convergence results of clip-SGD with heavy-tail noise are standard.

### Weaknesses

* The majority of the convergence results seem to be incremental, without any significant theoretical challenges highlighted by the authors compared to existing works.

* The numerical evaluation is significantly lacking, with performance evaluated only on a synthetically generated problem, using artificially injected gradient noise.

* The authors briefly mention adaptive methods, which are shown to perform well in practice, however, they do not consider any further comparison, either empirically or theoretically.

* The bound in Theorem 4.2 under bounded $p$-th moment is sub-optimal compared to the known lower-bounds in both convex and general non-convex scenarios.

---

> ### Author Rebuttal · Authors · 2026-03-28
>
> Our response to your constructive comments on this work is as follows:
>
> **For  Q1&W4:**
> - **The upper bound in Theorem 4.2 is tight:** **[Theorem 5.5, R1]** shows that there always exists a $(0, L_\nu, \nu)$-Hölder smooth and $G$-Lipschitz function $f$ such that, for sufficiently small $\varepsilon$, if the domain diameter satisfies $D \geq \Delta_1/\varepsilon$, then the sample complexity of SGD with $\eta = t^{-r}$ to achieve $E[|\nabla f(x)|] \leq \varepsilon$ is at least $\Omega(\varepsilon^{-\frac{2}{1-r}} + \varepsilon^{-\frac{p+\nu-2}{r(p-1)(\nu-1)}})$.  When $\nu = 2$, this reduces to the standard smooth case (a special case of weakly convex optimization) $\Omega (\varepsilon^{-\frac{2}{1-r}} + \varepsilon^{-\frac{p}{r(p-1)}})$, which matches the upper bound in Theorem 4.2 with $r=1/p$.
> - **The upper bound in Theorem 5.1 cannot be sharpened:** eq.(80) in Appendix D.7 shows that $(1/T)\sum_{t=1}^T E\|\nabla f_{1/2\rho}(x_t)\|^2 \leq O(T^{(1-p)/p})$.  Assume this upper bound can be improved to $O(T^{-r})$ with $r > (p-1)/p$. Consider a weakly convex function from **[Example 1 in 3.2, R2]**: $$f(x) = -x^2 + 1 \, x\in (-1,-0.5)$$ and $$f(x)=3(x+1)^2\, otherwise$$ $f$ is 2-weakly convex with $f_* = \inf f = 0$ and satisfies $f(x) - f_* \leq \text{dist}^2(0, g)\, g\in \partial f(x)$ in $R$.  Hence, by eq. (3) in line 198, we have $(1/T)\sum_{t=1}^T E[f(x_t)] - f_* \leq O(1/T^r)$.  Previous work shows that Clip-SGD for convex functions cannot converge faster than $\Omega(T^{(1-p)/p})$ in terms of $f(x) - f_*$, implying that $f$ would allow Clip-SGD to find its minimum even faster than any convex function. Such a contradiction reveals that the upper bound in Theorem 5.1 cannot be improved.
>
> **For Q2:**   We compare our results with some adaptive methods in heavy-tailed assumption
> | Method| Objective| Constrained domain| Complexity|
> |---|---|---|---|
> |AdaGrad-Norm [R3]| convex|Bounded|$O(\varepsilon^{-p/(p-1)})$|
> |Clip-AdaGrad-Norm [R4]|$L$-smooth| $R^d$|$O(\varepsilon^{-(3p-2)/(2p-2)})$|
> |Clip-AdaGrad-Norm [R4]|convex|$R^d$|$O(\varepsilon^{-p/(p-1)})$|
>
> When the noise is heavy-tailed, we point out that no existing result establishes the convergence of adaptive methods (with or without clipping) for weakly convex functions in bounded or unbounded constrained domains.
>
> **For the weakness**:
> - **W1:** We first explain the limitation of existing techniques in weakly convex analysis. Even in analyzing adaptive methods, the goal is to obtain inequality of the form: $E[f_{1/\hat{\rho}}(x_{t+1})|x_t] - f_{1/\hat{\rho}}(x_t)\leq -\Omega(\eta_t) |\nabla f_{1/\hat{\rho}}(x_t)|^2 + O(\eta_t^2)E[|\epsilon_t|^2|x_t] + O(\eta_t^2)G^2$, where $\epsilon_t$ is the difference between the chosen stochastic estimator and the true subgradient. However, $O(\eta_t^2)E[|\epsilon_t|^2|x_t]$ cannot be bounded when the gradient noises are $p$-BCM. We discuss this challenge in lines 239–267 (1st column) of the submission.  To address it, we avoid using the non-expansiveness of the projection at the very beginning and carefully apply _Young's Inequality_ to obtain Lemma 4.1.  To the best of our knowledge, Lemma 4.1 is new, and has not been studied in the weakly convex optimization community; Reviewer FzjF also confirmed this point. Second, for analyzing Clip-SGD, we construct two martingales $\eta_t\langle \epsilon^u_t, \hat{x_t}-x_t\rangle$ and $\eta_t^2(|\epsilon^u_t|^2-E_t|\epsilon|^2)$ (see eq. (65) ) which are not considered for vanilla SGD.  Our probabilistic analysis for the martingales is key to get $O(\varepsilon^{-2p/(p-1)})$ and does not appear in previous weakly convex literature. For the above points, we will add further discussion in the revision.
>
> - **W2 and W3** We understand your concerns about not evaluating on real data and models. we will include some realistic experiments experiments such as fine-tuning the language model in the revision.  To compare the non-adaptive methods with the adaptive methods empirically, we evaluated some methods (SGDM refers to SGD with momentum) on ResNet18 with the image dataset CIFAR10. All hyperparameters are either empirically fine-tuned or set to default parameters in PyTorch. The following table reports the average loss of 5 training sessions after 50 epochs. We will include more visual results in the revision.
>
>   | |SGDM|Clip-SGDM|Adam|Clip-Adam|
>   |---|---|---|---|---|
>   |Loss|$0.47\pm0.06$| $0.36\pm0.004$|$0.44\pm0.007$|$0.40\pm0.003$|
>   |Test(%)|$78.94\pm0.88$|$80.37\pm0.28$|$80.63\pm0.56$|$81.25\pm0.83$|
>
> From the table, although Adam performs better than SGDM, gradient clipping always improves performance in practice, which is consistent with previous work observations.
>
>  > [R1]Fatkhullin, et al. https://arxiv.org/abs/2508.04860v1
>
>  > [R2]Liao, et al. https://arxiv.org/abs/2312.16775v3
>
>  >[R3]Zijian. https://arxiv.org/abs/2508.07473v3
>
>  >[R4]Chenzhegov, et al. https://arxiv.org/abs/2406.04443v3

---

> > ### Author Rebuttal · Reviewer_nU1G · 2026-04-04
> >
> > I thank the authors for their responses. However, after reading the responses, I maintain my initial judgement. I still do not find the argument of novelty and technical challenges compelling, as heavy-tailed noise has been extensively studied at this point, across a myriad of settings. Further, the existing numerical evaluation remains weak. On the response to Q1 & W4, the authors consider non-smooth setting in their work, but compare to a lower bound from [R1] with a special case of Holder smooth with $\nu = 2$, which is different than their assumed setting, mixing concepts. All in all, my impression remains unchanged and I keep my initial score.
> >
> > ## **Update**
> >
> > I thank the authors for their further response and clarification. Having gone over it in details and after revisiting the original response, I can see further merits of the work. Therefore, I am increasing my original score.

---

> > > ### Author Response · Authors · 2026-04-04
> > >
> > > We appreciate your feedback, but we would like to reiterate that the **convergence analysis of SGD and clipped SGD under heavy-tailed noise in weakly convex settings is novel**. Although heavy-tailed noise has been widely studied under convex or smooth conditions, **no existing work analyzes algorithm convergence in weakly convex settings, particularly when gradient clipping and normalization are not used**.
> > >
> > > We point out that **weakly convex functions include all convex functions and $L$-smooth functions, not just nonsmooth ones.**
> > >
> > >  For **Q1:**
> > >  **The upper bound in Theorem 4.2 is indeed tight:**  Recall the definition of upper complexity: $O(H,\varepsilon, A) = \sup_{f\in H} \inf_T E[|\nabla f(x_{T}')|\leq \varepsilon]$, where $A$ is the algorithm instance (e.g., vanilla SGD), $x_{T}'$ is the output of $A$ after $T$ iterations and $H$ is the function class we studyed. Since $H_1({L-\rm smooth\ functions})\subset H_2({\rm weakly\ convex\ functions})$, we have $\Omega(H_1,\varepsilon, A)\leq O(H_1,\varepsilon, A)\leq O(H_2,\varepsilon, A)$.
> > > For the convergence measure, note that $|\nabla f(x)| \leq |\nabla f_{1/\hat{\rho}}(x)|$, so the upper complexity measured by the gradient of the envelope function recovers the complexity measured by the gradient of the original function.
> > > When $\nu=2$, the lower bound for vanilla SGD in [Theorem 5.5, R1] under the heavy-tailed setting reduces to the $L$-smooth case: $\Omega(\varepsilon^{-\frac{2p}{p-1}})$ with step-size $\eta = t^{-1/p}$ and constrained domain of diameter $D\geq\Delta_1/\varepsilon$.
> > >
> > > > Importantly, for $\nu=2$, **the hard instance they proposed in [Proposition 5.7, R1] to prove the lower bound** is reduces to following
> > > $
> > > F(x) = (L/2)\|x\|^2,\ \|x\|\leq (G/2L),\quad F(x) = (G/2)\|x\| - G^2/8L,\ \|x\|>G/2L
> > > $
> > > , **then $F(x)$ is convex (The Hessian matrix of $F(x)$ is non negative definite) and $L$-smooth surely $F(x)$ is in the weakly convex function class**
> > >
> > > Using the same step-size and consider a general bounded domain, we have shown that the upper bound of vanilla SGD in the weakly convex setting is $O(\varepsilon^{-\frac{2p}{p-1}})$, matching the lower bound. Hence, the upper bound in Theorem 4.2 of this paper is tight.
> > >
> > > **The upper bound in Theorem 5.1 is hard to be sharpened:** The key idea to show this is to construct a nonconvex but weakly convex function $f$ that satisfies the PL condition [Example 1 in 3.2, R2]. If the upper bound could be improved, then the rate of Clip-SGD for finding an optimal solution of $f$ on $R^d$ would be even faster than that for all convex functions. This reveals that the upper bound in Theorem 5.1 is hard to be sharpened for general weakly convex functions
> > >
> > > **For $W4$**:
> > >
> > >  > The bound in Theorem 4.2 under bounded
> > > -th moment is sub-optimal compared to the known lower-bounds in both convex and general non-convex scenarios.
> > >
> > > To the best of our knowledge, the most known lower bounds under heavy-tailed noise are for normalized SGD, Clip-SGD, or their variants, typically under convex or (generalized) smooth settings with unconstrained domains. [R1] provides the first and only lower bound for vanilla SGD in Hölder smooth settings; when $\nu=2$, it reduces to the $L$-smooth setting. As explained in our response to Q1, this lower bound can be used to demonstrate the tightness of the upper bound in Theorem 4.2.
> > >
> > >
> > > [R1]Fatkhullin, et al. https://arxiv.org/abs/2508.04860v1
> > >
> > > [R2]Liao, et al. https://arxiv.org/abs/2312.16775v3

---

### Official Review · Reviewer_FzjF · 2026-03-14

**Soundness:** 4
**Presentation:** 4
**Significance:** 3
**Originality:** 3
**Overall Recommendation:** 5
**Confidence:** 3

**Summary:**

The authors minimize the objective $\min_{x\in \mathcal{X}} f(x)$, where $f(x) = \mathbb{E}_{\xi}[f(x,\xi)]$ and the function $f$ is a  $\rho$-weakly convex function ( Assumption 3.1) and is $G$-Lipschitz (Assumption 3.2) on a convex closed domain $\mathcal{X}$.

The authors assume that they have access to unbiased stochastic subgradients, $g(x, \xi)$ such that $\mathbb{E}[g(x, \xi)] \in \partial f(x)$. However, these stochastic subgradients are heavy-tailed, i.e., $\mathbb{E}[||g(x, \xi) - \mathbb{E}[g(x, \xi)]||^2] =\infty$, or the gradient noise variance is unbounded. The authors consider two classes of functions with heavy tails --
i) ($p$-Bounded Central Moment, Assumption 1.1) where the $p^{th}$ central moment of the noise is bounded by $\sigma^p$ for some $p\in(1,2]$, ii) ($\sigma$-SubWeibull($\theta$), Assumption 1.2), where the noise in the subgradient is sub-Weibull with parameter $\theta$. Note that sub-Weibull distributions are a generalization of sub-Gaussian ($\theta = \frac{1}{2}$) and sub-Exponential distributions ($\theta=1$), and weaker than Assumption 1.1.

Existing analysis for weakly-convex function requires bounded variance of stochastic gradients or sub-Gaussian noise, while existing analysis of heavy-tailed noise requires convexity or smoothness alongwith algorithmic modifications like clipping or normalization. Therefore, the case of weakly convex optimization under heavy tail noise has not been tackled before. The authors prove 3 important theoretical results for this problem. The main theoretical results that authors provide are the following. All theoretical results are on the time-averaged gradient norm of the Moreau envelope, which is a common optimality measure for weakly convex objectives without any growth conditions, like KL/PL/Sharpness.

1. Under $p$-BCM, vanilla SGD converges in $O(G^2 \epsilon^{-4} + \sigma^{\frac{p}{p-1}}\epsilon^{-\frac{2p}{p-1}})$ iterations to $\epsilon$ expected error when the domain $\mathcal{X}$ is bounded (Theorem 4.2).
2. Under sub-Weibull noises, with probability $1-\delta$, vanilla SGD converges in $O(\epsilon^{-4} poly\log(\frac{1}{\delta}, \frac{1}{\epsilon}))$ iterations for even unbounded domain(Theorem 4.4).
3. Under $p$-BCM, with the knowledge of $p$, with probability $1-\delta$, clip-SGD converges in $O((G^2\epsilon^{-4} + \sigma^{\frac{p}{p-1}}\epsilon^{-\frac{2p}{p-1}})\log(\frac{1}{\delta}))$ iterations when domain is bounded (Theorem 5.1).

For Theorem 4.4, they don't require the knowledge of $p$, however for Theorems 4.2 and 5.1, they require the knowledge of $p$.

They further verify their theoretical results by numerical experiments on the a linear regression problem with $\ell_1$ loss with heavy-tailed gradient noise (Fig 1 and 2).

**Compliance With Llm Reviewing Policy:**

Affirmed.

**Final Justification:**

My final recommendation is acceptance. I did not have any major concerns about the paper. The authors' rebuttal has addressed my additional questions.
I believe that the analysis for weakly convex functions under heavy-tailed noises is novel and an important contribution for the optimization community.

**Key Questions For Authors:**

Please address the questions in the weaknesses.

**Limitations:**

Yes. The authors discuss this in their Conclusion.

**Strengths And Weaknesses:**

## Strengths --
- **Novel Theoretical Analysis**: Their theoretical analysis is novel. Specifically, the descent lemma for weakly-convex functions has been modified, using a Young's-inequality like form to handle the case of $p^{th}$ moments of noise being bounded (See Lemma 4.1 and Eq (35) page 17). To the best of my knowledge, this modification for vanilla SGD has not been studied, and existing works for non-convex case that recover such bounds crucially use smoothness or algorithmic modifications like clipping or normalization.

- **Tight analysis**: The authors recover known results for weakly-convex optimization if the heavy-tailed noise actually becomes a bounded variance noise. Hence, the analysis seems tight.

- Presentation: The presentation of the paper is very good. It is easy to read and understand, barring a few typos. Explanation of the proof sketch and why existing analysis for SGD on weakly-convex functions doesn't work was very helpful in particular.

- **Thorough Related Works**: Section 2, Appendix  A, and in particular Table 1 are very thorough. Table 1 allows one to easily understand the scope of novelty of the paper and compare it to previous results in weakly convex optimization. I'd recommend moving Table 1 to the main text for the final version if possible.


## Weaknesses --
I don't think the paper has any major weaknesses.
- **Typos** :
   1. Line 014 2nd column, "objecitve" -> "objective"
   2. Page 3, Point 1. (a) : $G$ hasn't been defined yet.
- **Discussion of Tightness**: It looks like the $O(\epsilon^{-\frac{2p}{p-1}})$, the complexity under $p$-BCM for Vanilla SGD is same for Holder-Smooth(Fatkhullin et al 2025) and weakly-convex functions. Note that for bounded variance we know that this is the case, but this paper shows that this is still true for heavy tails. I think this hasn't been explicitly mentioned by the authors. Further, we don't know if the bounds for SGD are tight or not. I could not find a lower bound for weakly-convex or non-convex, non-smooth but Lipschitz gradients under heavy-tailed noise. However, I feel such a lower bound might be a good direction for future work. Further, the parameter-free results being weaker suggests that there is a gap between the optimal bounds for the parameter-free case and the case when we know all parameters. For lower bounds, it might be worthwhile to construct functions where Eq (5) is tight.
- **Transition from $p$-BCM to sub-Weibull and from SGD to clipped SGD:** The reason for transitioning from BCM to sub-Weibull conditions, in my understanding, is to go from weaker convergence in expectation to stronger high probability convergence guarantees, while keeping the algorithm SGD. The transition from SGD to clipped-SGD is also for the same reason, but it changes the algorithm while keeping the noise assumption BCM. I think these transitions and the reasons for it have not been particularly well explained. For instance, is sub-Weibull the minimum relaxation of BCM which gives high probability guarantees for SGD? Is clipped SGD the minimum algorithmic change to achieve high-probability guarantees for BCM. Additionally, while Lemma 4.1 is very good for understanding Theorem 4.2, explanation of how high-probability bounds are not obtained for Vanilla SGD with BCM should be provided. That would help explain the need for Theorem 4.4 and Theorem 5.1. Further, a smaller proof sketch of the key argument for these proofs in the main text would also be good.
- **How convexity and smoothness and algorithmic modifications bypass the need for Young's inequality in Lemma 4.1:** Following up on the previous point, the authors state several times that these additional conditions achieve optimal rates for heavy tails, but they don't provide an intuitive explanation for why this is the case.

- **More realistic experiments:** In the introduction, the authors mention several real problems that satisfy heavy tails. If the authors could find a more realistic example of a data distribution and model that are weakly convex and heavy-tailed, with a tail index they can estimate, adding it to the experiments would improve the paper's practical applicability. They don't need to do this necessarily, but it would be a great addition.
- **Table for comparison of heavy-tailed noise under different conditions on the objective:** If the authors could have a table similar to Table 1 but for heavy-tailed noise, varying the assumptions on objective and algorithm, like strong-convexity/convexity, and normalization/clipping, it would make Section 2 much smaller but more easier to understand.

---

> ### Author Rebuttal · Authors · 2026-03-25
>
> Our response to your constructive comments on this work is as follows:
>
> > [R1] Fatkhullin, et al. https://arxiv.org/abs/2508.04860v1
>
> - **Typos:** $G$ in Page.3 1(a) is actually the uniform bound of the norm of subgradients in Assumption 3.2.
>
> - **Discussion of Tightness:** **[R1]** indeed achieves the same sample complexity upper bound as ours for weakly convex functions under Hölder smoothness. We will explicitly mention this in the revision. In fact, **we provide tight complexity upper bound in Theorem 4.2 for the weakly convex functions**, since weakly convex functions include the smooth functions and thus inherit the same lower bound $\Omega(\varepsilon^{-\frac{2p}{p-1}})$ with $\eta=t^{-1/p}$ established in **[R1, Theorem 5.5]** for smooth functions (with $\nu=2$ corresponding to the smooth case).  However, no existing work establishes lower bound complexity for SGD in weakly convex problems under unconstrained optimization, even in the bounded-variance setting. At present, we do not know whether the rate $O(\varepsilon^{-\frac{2p}{p-1}})$ remains achievable when $p$ is unknown, and we leave this as an interesting direction for future research. For the above points, we will add further discussion in the revision. Moreover, thanks for your suggestion on the presentation. We will include a discussion on why vanilla SGD is hard to achieve high-probability convergence under $p$-BCM noises, along with a proof sketch, in the revision.
>
> - **Transition from $p$-BCM to sub-Weibull and from SGD to clipped SGD:** First, we explain the motivation for transitioning from $p$-BCM to sub-Weibullassumptions. Under the bounded variance condition, a standard approach in high-probabilityanalyses of SGD is to equivalently reformulate it as sub-Gaussian noise, which facilitates the application ofFreedman-type concentration inequalities. Analogously, under the $p$-BCMcondition, we naturally consider the sub-Gaussian type transformation $E_{\xi}[\exp( \| \nabla f(x)-g(x,\xi) \|^p/ \sigma^p )]\leq e$, which corresponds (up to constants) to assuming sub-Weibull gradient noise. However, this transformation is not equivalent when $p<2$: the sub-Weibull condition is strictly stronger than $p$-BCM. Second, the motivation behind transitioning from SGD to clipped SGD is to achieve high-probability convergence in unbounded domains. Such a modification is necessary. For example, let the constraint domain be $R$, and $f=E_\xi[(1/2)x^2+\xi]$, where $\xi$ is zero-mean noise with infinite variance and bounded $p$-th moment. After one step of SGD starting from $x_1 =0$, we get $E[f(x_2)]=1/2E[|f^\prime (x_2)|^2]=1/2E[|\eta_1\xi|^2]= +\infty$, showing that vanilla SGD is hard to be guaranteed convergence. Regarding an interestingand open question you proposed—whether there exist heavy-tailed noise models between $p$-BCM and sub-Weibull that still preserve the convergence of vanilla SGD—we believe that addressing it likely requires more refined structural assumptions on the gradient noise beyond moment conditions alone.
>
> - **How convexity and smoothness and algorithmic modifications bypass the need for Young's inequality in Lemma 4.1:** The standard convergence analysis of (projected) SGD in the convex setting studies the contraction of $|x_t - x_*|^2$, where $x_*$ is a global minimizer. Under the $p$-BCM gradient noise assumption, **[R1]** shows that one can instead analyze the contraction of $|x_t - x^*|^p$, so the classical framework still applies (for the smooth setting, they adopt $E[\|\nabla f(x_t)\|^p]$ as convergence measure). However, for weakly convex functions, directly following prior analyses with measure $E[\|\nabla f_{1/\hat{\rho}}(x_t)\|^2]$ leads to $f_{1/\hat{\rho}}(x_{t+1})-f_{1/\hat{\rho}}(x_{t})\leq 2\eta_t\langle g_t, \hat{x}_t-x_t \rangle + \eta_t^2|g_t|^2$, where the $p$-BCM assumption prevents effective control of $\eta_t^2|g_t|^2$. To address this issue, we refine the classical SGD analysis in the weakly convex setting. In particular, for Clip-SGD, gradient clipping ensures that $E[|g^{\rm clip}_t-\nabla f(x)|^2]$ is bounded (see Lemma D.1 in Appendix D), thereby mitigating the heavy-tailed noise and bypassing the need for Young's inequality in Lemma 4.1.
>
> - **More realistic experiments & Comparsion table :** For the experiments, we understand your concern on the lack of realistic experiment. However, it is difficult to determine the type of heavy-tailed distribution of gradient noises in practice, let alone estimate its tail parameter, when training on real-world data. In such cases, hyperparameters are typically tuned empirically, which may weaken the validation of theoretical insights—especially since we do not propose any new algorithms. Regarding the suggested comparison table, we agree that it is valuable. Due to space limitations, we do not include it here, but we will incorporate comparisons with results under other settings in the revision.

---

> > ### Author Rebuttal · Reviewer_FzjF · 2026-03-31
> >
> > The rebuttal addresses all my concerns. I don't have any additional questions. I would recommend including some of the explanation in this rebuttal in the final version.

---

> > > ### Author Response · Authors · 2026-04-06
> > >
> > > We appreciate your rebuttal  acknowledgement, and we agree with that incorporate some of the discussion, particularly regarding tightness, the motivations on the transition and some proof techniques, into the final version.

---

### Decision · Program_Chairs · 2026-04-30

**Decision:**

Accept (regular)

**Comment:**

This paper studies stochastic gradient methods under heavy-tailed noises for weakly convex optimization, providing convergence analyses for vanilla SGD and Clip-SGD under p-BCM and sub-Weibull noise assumptions.

The submission received four mixed reviews with scores ranging from 3 to 5, with two in favor of acceptance and two leaning weak rejection. Reviewers acknowledge the novel theoretical analysis using Young’s inequality, rigorous proofs, and clear presentation with comprehensive comparisons to prior work. Concerns include limited novelty against a prior arXiv preprint paper, insufficient practical experiments on modern models, and minor presentation issues. The authors provided thorough rebuttals that resolved most technical doubts and clarified novelty and experimental limitations. Following a sanity check, we confirm that the arXiv preprint mentioned by Reviewer YzVg does not violate ICML’s policy on concurrent submissions, and the originality of the present work remains fully valid.

Overall, this work offers solid and novel contributions to weakly convex optimization under heavy-tailed noise. We hereby recommend weak accept. The authors are strongly encouraged to incorporate rebuttal discussions, enhance real-world experiments, and fix presentation issues in the revised manuscript.